# Blink of an eye: a simple theory for feature localization in generative models

Marvin Li[1]  Aayush Karan[2]  Sitan Chen[2]

## Abstract

Large language models can exhibit unexpected behavior in the blink of an eye. In a recent computer use demo, a language model switched from coding to Googling pictures of Yellowstone, and these sudden shifts in behavior have also been observed in reasoning patterns and jailbreaks. This phenomenon is not unique to autoregressive models: in diffusion models, key features of the final output are decided in narrow "critical windows" of the generation process. In this work we develop a simple, unifying theory to explain this phenomenon. Using the formalism of stochastic localization for generative models, we show that it emerges generically as the generation process localizes to a sub-population of the distribution it models. While critical windows have been studied at length in diffusion models, existing theory heavily relies on strong distributional assumptions and the particulars of Gaussian diffusion. In contrast to existing work our theory (1) applies to autoregressive and diffusion models; (2) makes very few distributional assumptions; (3) quantitatively improves previous bounds even when specialized to diffusions; and (4) requires basic mathematical tools. Finally, we validate our predictions empirically for LLMs and find that critical windows often coincide with failures in problem solving for various math and reasoning benchmarks.

## 1. Introduction

In large language models (LLMs), undesirable behavior can often emerge very suddenly. For example,

- Claude transitioned from coding to browsing pictures of Yellowstone while using a computer (Anthropic, 2024).

[1]Harvard College, Cambridge, MA, USA [2]John A. Paulson School of Engineering and Applied Sciences, Harvard University, Cambridge, MA, USA. Correspondence to: Marvin Li <marvinli@college.harvard.edu>.

*Proceedings of the 42nd International Conference on Machine Learning*, Vancouver, Canada. PMLR 267, 2025. Copyright 2025 by the author(s).

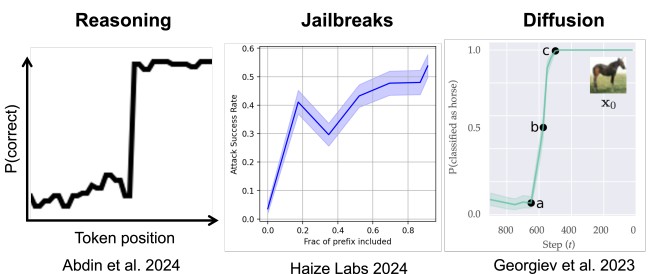

*Figure 1.* Examples of critical windows for different data modalities and samplers, including reasoning (Abdin et al., 2024; Qi et al., 2024) and certain jailbreaks (Haize Labs, 2024b) for language models and image class for diffusion models (Georgiev et al., 2023).

- The Phi-4 team reported that the probability of correctly answering a math problem can plummet with a single token (Abdin et al., 2024; Lin et al., 2024).
- Gemini abruptly threatened a student who was using it to study (Gemini, 2024).
- Llama models can be jailbroken by manipulating the first handful of generated tokens (Qi et al., 2024; Haize Labs, 2024b).

These abrupt shifts are not unique to autoregressive models. In diffusion models, it has been observed that certain properties like the presence of an object in the background or the image class emerge in narrow time intervals, sometimes called *critical windows*, of the generation process (Ho et al., 2020; Meng et al., 2022; Choi et al., 2022; Raya & Ambrogioni, 2023; Georgiev et al., 2023; Sclocchi et al., 2024; 2025; Biroli et al., 2024; Li & Chen, 2024).

Critical windows, more broadly characterizable as *a few steps of the sampling procedure* during which features of the final output appear, arise in many different generative models and data modalities (Figure 1). They are extremely useful from an interpretability perspective as they represent the steps of the sampler responsible for a given property of the output (Georgiev et al., 2023; Qi et al., 2024), and have also been used to provide richer stepwise rewards for preference optimization and finetuning (Abdin et al., 2024; Lin et al., 2024; Qi et al., 2024). As the applications of generative models proliferate, it is crucial from interpretability, safety, and capability perspectives to understand how and

why these critical windows emerge.

Recently, this phenomenon has received significant attention within the theoretical literature on diffusion models (Raya & Ambrogioni, 2023; Sclocchi et al., 2024; 2025; Biroli et al., 2024; Li & Chen, 2024). While existing works do offer predictive theory in the diffusions setting, they either (A) make strong distributional assumptions or (B) rely heavily on the particulars of diffusion, which do not straightforwardly extend to autoregressive models.

Works in the former category carry out non-rigorous statistical physics calculations tailored to specific toy models of data like mixtures of Gaussians or context-free grammars with random production rules (Sclocchi et al., 2025; 2024; Biroli et al., 2024; Raya & Ambrogioni, 2023). Works in the latter category derive rigorous bounds in settings without explicit parametric structure, e.g. mixtures of strongly log-concave distributions (Li & Chen, 2024), but they rely on tools like Girsanov's theorem which are specific to Gaussian diffusion. Additionally, the bounds in the latter are generally cruder, losing dimension-dependent factors. We ask:

*Is there a simple, general theory that can explain critical windows across all generative modeling paradigms and data modalities?*

### 1.1. Our contributions

In this work, we develop a simple theoretical framework that characterizes critical windows in both diffusion models and autoregressive models. Our theory is fully rigorous and show that such windows arise generically when the model localizes from a larger sub-population to a smaller one, distilling the phenomenon of critical windows to very general facts about the data distribution. In particular, we show that the critical window is parameterized by the last step of the generative model at which the larger and smaller sub-populations are indistinguishable and the first step of the generative model at which that smaller and larger sub-populations have negligible overlap between each other. We will formalize these notions in Section 3.

The key insight of our theory is to apply to the formalism of *stochastic localization samplers* (see Section 2.1 for a formal description) to understand critical windows (Montanari, 2023b; Chen & Eldan, 2022). Roughly speaking, a stochastic localization scheme is any generative model given by a time-reversal of a Markovian degradation process which takes a sample from the target distribution and generates progressively less informative "observations" of it. In diffusion models, the degradation is a convolution of the original sample with larger and larger amounts of Gaussian noise. In autoregressive models, the degradation is the masking of entries from right to left. Importantly, our theory does not use anything about the specific structure of the sampler

beyond the Markovianity of the observation process. Below we highlight our main contributions:

1. **Generality**: In comparison to existing work, our theory (Theorem 3.1) makes very few distributional assumptions and requires no statistical physics or stochastic calculus machinery, relying only on simple mathematical tools. The simplicity and generality of our theory yield concrete improvements to rigorously characterizing the location of critical windows for well-studied models of data. For example, in contrast to (Li & Chen, 2024), who could only identify an analogue to Theorem 3.1 with bounds that grow at a polynomial factor with dimension, our improved theory allows us to obviate this dimensional dependence and replace it with a constant.

2. **Diverse instantiations**: Another blessing of the generality of our framework is that we can characterize the location of critical windows for many different models of data and generative models, while previous works were restricted to one particular form of data or model. To illustrate the flexibility of our bounds, we explicitly compute the locations and widths of these windows for different generative models and data modalities (Section 4). One such example we provide elucidates a new connection between critical windows for in-context learning and the all-or-nothing phenomenon in statistical inference.

3. **Insights into hierarchical data**: We instantiate our bounds for hierarchically structured models of data, significantly generalizing results of (Li & Chen, 2024) which only applied to diffusions and Gaussian mixtures (Section 5). This allows us to show that the hierarchy for a generative model may resemble the hierarchy of the true data generating process if both come from the same kind of sampler, but in general may differ. We also use our theory to argue that autoregressive models can support deeper hierarchies than diffusion models.

4. **Experimental results**: Finally, we show that our theory makes accurate predictions for the location of critical windows for LLMs in a toy synthetic setting. We also empirically demonstrate critical windows for generations from LLAMA-3.1-8B-Instruct, Phi-3-7B-Instruct, and Qwen-2.5-7B-Instruct on 7 different math and reasoning benchmarks. Concurrently with (Abdin et al., 2024; Lin et al., 2024), we observe that critical windows occur during important mistakes in the reasoning patterns of LLMs. [1]

Our theory provides valuable insights for practitioners. For instance, in Example 4.3 we provide a model for critical windows in jailbreaks and the Yellowstone example (Anthropic, 2024; Qi et al., 2024), and argue that training on correc-

---

[1] The code to reproduce the experiments can be found at `https://github.com/marvinli-harvard/critical-windows-lm`.

tions from critical windows can enable models to recover from these 'bad' modes of behavior. This provides rigorous theoretical justification for (Qi et al., 2024)'s approach for deepening safety alignment through finetuning.

### 1.2. Related work

We briefly overview some related work here and defer our discussion of other relevant literature to Appendix A.

**Theory of critical windows in diffusion.** Several recent works have studied critical windows in the context of diffusion models, using either statistical physics methods (Raya & Ambrogioni, 2023; Sclocchi et al., 2024; 2025; Biroli et al., 2024) or Girsanov's theorem (Li & Chen, 2024). The statistical physics papers assume an explicit functional form for the data and use accurate and non-rigorous statistical physics methods to compute critical windows. For instance, (Biroli et al., 2024) computes the critical time at which the reverse process specializes to one component for a mixture of two spherical Gaussians using a Landau-type perturbative calculation, and (Sclocchi et al., 2025; 2024) passed through a mean-field approximation to compute the critical windows for a *random hierarchy model* (Petrini et al., 2023), a multi-level context-free grammar with random production rules. Our work is most similar to (Li & Chen, 2024), which derives rigorous, non-asymptotic bounds analogous to our Theorem 3.1 for mixtures of log-concave distributions with Girsanov's theorem (Chen et al., 2023).

In contrast to existing work, our theory applies to all localization-based samplers, including diffusion and autoregressive language models, and imposes no functional form or log-concavity assumptions on the distribution. We also improve upon the main theorem of (Li & Chen, 2024) by obtaining dimension-independent error bounds. Using our improved theorem, we can extend the definition of hierarchy of critical windows from (Li & Chen, 2024) to all localization-based samplers and, for continuous diffusions, to distributions beyond mixtures of Gaussians.

**Forward-reverse experiment.** Here we study critical windows with the forward-reverse experiment, where we *noise and denoise* samples with a given attribute to understand critical windows. This was also explored in (Li & Chen, 2024; Sclocchi et al., 2025; 2024). This approach is very similar to the framework in which one imagines re-running the reverse process at an intermediate point $Y_t$ (Georgiev et al., 2023; Biroli et al., 2024; Raya & Ambrogioni, 2023). Both perspectives provide rigorous frameworks to understand critical windows, and in the case where the forward process is deterministic, i.e. autoregressive language models, these frameworks are equivalent.

**Stochastic localization.** (El Alaoui et al., 2022; Montanari & Wu, 2023; Alaoui et al., 2023; Montanari, 2023a; Huang et al., 2024) applied Eldan's stochastic localization method (Eldan, 2013; 2020) to develop new sampling algorithms for distributions inspired by statistical physics. Our work applies the stochastic localization framework (Montanari, 2023b) to understand an empirical phenomenon appearing among different localization-based samplers widely used in practice.

## 2. Technical preliminaries

**Probability notation.** Given distributions $P, Q$ defined on $(\Omega, \mathcal{F})$ with a base measure $\mu$, the *total variation distance* is defined as $\mathrm{TV}(P, Q) \triangleq \frac{1}{2} \int |dP - dQ| d\mu$. For random variables $X, Y$, we will also use $\mathrm{TV}(X, Y)$ as shorthand to denote the TV of the measures of $X, Y$. Let $\mathrm{supp}(P) = \{x \in \Omega | dP(x) > 0\}$ denote the support. We will also use the following well-known relationship.

**Lemma 2.1.** *For probability measures $P, Q$,* $\mathbb{E}_{x \sim P} \left[ \frac{dQ}{dP + dQ} \right] \leq \frac{1}{2} \sqrt{1 - \mathrm{TV}^2(P, Q)}$.

To study critical windows in diffusion and autoregressive models, we consider a *forward-reverse* experiment. A forward-reverse experiment considers the amount of "noise" one would need to add to a generation so that running the generative model starting from the noised generation would still yield a sample with the same feature. For a diffusion model, this could mean taking an image of a cat, adding Gaussian noise, and resampling to see if the result is still a cat. For a language model, it could mean truncating a story about a cat and resampling to check if the story remains about a cat. Now, we will use the language of stochastic localization to place these analogous experiments for diffusion and language models within the same framework.

### 2.1. Stochastic localization samplers

We formally define the framework for stochastic localization samplers, following (Montanari, 2023b). Let $X \sim p$ be a random variable over $\mathbb{R}^d$.[2] We consider a sequence of random variables $(Y_t)_{t \in \mathbf{I}}$ with a compact index set $\mathbf{I} \subset [0, \infty) \bigcup \{\infty\}$. As $t$ increases, $Y_t$ becomes *less informative* and *degrades* the original information about $X$ (Definition 2.2). As in (Montanari, 2023b), we will only consider *complete* observation processes, where information about the path $(Y_t)_{t \in \mathbf{I}}$ uniquely identifies $X$: for any measurable set $A \subset \mathbb{R}^n$, we require $P(X \in A | (Y_t)_{t \in \mathbf{I}}) \in \{0, 1\}$. For the sake of simplicity, we will assume $0, \infty \in \mathbf{I}$ and $Y_\infty$ is totally uninformative about $X$.

**Definition 2.2.** $(Y_t)_{t \in \mathbf{I}}$ is an *observation process* with

---

[2]These definitions are easily carried over to the setting where $X$ lives in a discrete space.

respect to $X$ if for any positive integer $k$ and sequence $t_1 < t_2 < \cdots < t_k \in \mathbf{I}$, the sequence $X \to Y_{t_1} \to Y_{t_2} \to \cdots \to Y_{t_k}$ forms a Markov chain.

Because $X \to Y_{t_1} \to \cdots \to Y_{t_k}$ is a Markov chain, its reverse $Y_{t_k} \to \cdots \to Y_{t_1} \to X$ is also a Markov chain. To any such observation process one can thus associate a generative model as follows:

**Definition 2.3.** Given observation process $(Y_t)_{t \in \mathbf{I}}$ and times $t_1 < \cdots < t_m = \infty$ in $\mathbf{I}$, the associated *stochastic localization sampler* is the algorithm that generates a sample for $X$ by first sampling $Y_{t_m}$ and then, for $k = m-1, m-2, \ldots, 0$, sampling from the posterior on $Y_{t_k}$ conditioned on $Y_{t_{k+1}}$ by taking one step in the reverse Markov chain above, and finally sampling $X$ conditioned on $Y_{t_0}$.

In Appendix B, we formally verify that diffusion and autoregressive models are special cases of this framework. In practice, one does not have access to the true posteriors of the data distribution and must learn approximations to the posterior from data. This issue of learning the true distribution is orthogonal to our work, and thus we define $X \sim p$ to be the sampler's distribution. Furthermore, it is more natural to study the sampler's distribution for applications such as interpretability or jailbreaks.

**Features, mixtures, and sub-mixtures.** To capture the notion of a feature of the generation, we assume that the distribution $X \sim p$ is a *mixture model*. Consider a discrete set $\Theta = \{\theta_1, \ldots, \theta_K\}$ with non-negative weights $w_1, \ldots, w_K$ summing to 1. Each $\theta_i \in \Theta$ is associated with a probability density function $p^{\theta_i} : \mathbb{R}^n \to \mathbb{R}^{\geq 0}$. To generate a sample $X \sim p$, we first draw $\theta \sim \mathrm{Cat}(\Theta, \{w_i\}_{i=1}^K)$ and return $X \sim p^\theta$. This yields an overall density of $p \triangleq \sum_{\theta \in \Theta} w_\theta p^\theta$. For any non-empty $S \subset \Theta$, we also define the sub-mixture $p^S$ by $p^S \triangleq \sum_{\theta \in S} \frac{w_\theta}{\sum_{\phi \in S} w_\phi} p^\theta$.

*Remark* 2.4. Note that the definition of $\Theta$ is extremely flexible and can be tailored to the particular data modality or task. For example, $\Theta$ could be $\{\mathbf{cat}, \mathbf{dog}\}$ for image diffusion models; $\{\mathbf{right}, \mathbf{wrong}\}$ for math and reasoning tasks; $\{\mathbf{unsafe}, \mathbf{safe}\}$ for jailbreaks. The ability to attach these labels and partition into subpopulations is broadly applicable to the datasets, which we also make use of in our experiments.

Here we study a *family of observation processes* corresponding to observation processes for different initial distributions of $X \sim p^S$ for $S \subset \Theta$. To ensure that we can meaningfully compare the observation processes within this family, we will assume that the *degradation procedure is fixed*. To formalize this intuition, we borrow the language from diffusion models of a forward process, which degrades $X$, and a reverse process, which takes a degraded $Y_t$ and produces $X$.

## 2.2. Forward-reverse experiment

Now we describe the general formalism under which we will study critical windows. Fixing some $t \in \mathbf{I}$ and $S \subset \Theta$, we start with some $X \sim p^S$, sample $Y_t | X$ from the observation process conditioning on $X$, and finally take $X' | Y_t$ from the stochastic localization sampler conditioning on $Y_t$. This can be understood as a generalization of the *forward-reverse* experiment in diffusions, originally studied in (Sclocchi et al., 2025; 2024; Li & Chen, 2024), to arbitrary stochastic localization samplers.

**Forward process.** For any $t \in \mathbf{I}$, define the forward Markov transition kernel $P_t^\rightarrow(A|X) = P(Y_t \in A|X)$. Note the forward Markov transition kernel does not depend on the distribution of $X$. The fact that the forward process is agnostic to the specifics of the original distribution is shared by the most widely used stochastic localization samplers. For example, in diffusion and flow-matching models, the forward transition is a convolution of $X$ with a Gaussian; in autoregressive language models, it is masking of the last remaining token in the sequence.

For any $t \in \mathbf{I}$ and $S \subset \Theta$, we let $p_t^S$ denote the law of $Y_t^S$, where we sample $X^S \sim p^S$ and then sample $Y_t^S \sim P_t^\rightarrow(\cdot|X^S)$. We omit the $\Theta$ in $p_t^\Theta$.

**Reverse process.** For any $t \in \mathbf{I}$ and initial distribution $X \sim p$, we define the posterior of $X$ given $Y_t$ by $P^\leftarrow(A|Y_t) = P_{X \sim p}(X \in A|Y_t)$, that is, the distribution of $X$ given by starting the sampling process at $t \in \mathbf{I}$ and $Y_t$ instead of $\infty$ and $Y_\infty$. We will also use this notation for the probability density.

Now, we are ready to describe the main forward-reverse experiment that we will study throughout the paper.

**Definition 2.5** (Forward-reverse experiment (Sclocchi et al., 2025; 2024; Li & Chen, 2024))**.** For nonempty $S \subset \Theta$ and $\widehat{T} \in \mathbf{I}$, let $p^{S,\widehat{T}}$ be the distribution of $X^{S,\widehat{T}}$ defined by the following procedure:[3]

1. Sample $Y_{\widehat{T}}^S \sim p_{\widehat{T}}^S$ — i.e. run the forward process for time $\widehat{T}$ starting at the sub-mixture $p^S$.

2. Sample $X^{S,\widehat{T}} \sim P^\leftarrow(\cdot|Y_{\widehat{T}}^S)$ — i.e. run the reverse process starting at $Y_{\widehat{T}}^S$ to sample from the posterior on $X$.

We emphasize that in the second step, we run the reverse process with the prior on $X$ given by the *entire distribution* $p$ rather than the sub-mixture $p^S$. If we did the latter, the marginal distribution of the result would simply be $p^S$. Instead, the marginal distribution of $X^{S,\widehat{T}}$ is some distribution whose relation to $p$ and sub-mixtures thereof is *a priori* unclear. Intuitively, as $\widehat{T} \to 0$, this distribution converges

---

[3]Note that this equips $2^\Theta$ with the structure of a poset, i.e. $A \subset B$ if and only if there exists some $t \in \mathbf{I}$ such that running the forward-reverse experiment up to $t$ from $p^A$ yields $p^B$.

to $p^S$, and as $\widehat{T} \to \infty$, this distribution converges to $p$. The essence of our work is to understand the transition between these two regimes as one varies $\widehat{T}$.

### 2.2.1. INSTANTIATION FOR LANGUAGE MODELS

Consider an autoregressive language model, which produces stories of cats or dogs. We have a discrete set of tokens $\mathcal{A}$ and $X \in \mathcal{A}^T$ representing length-$T$ sequences. Letting $\mathbf{I} = \{0, 1, 2 \ldots, T\}$, the *observation process* is defined with $Y_i \in \mathcal{A}^{T-i}$, $Y_0 \triangleq X$, and $Y_i$ being the first $T - i$ elements of $Y_{i-1}$ for $i = 1, \ldots, T$. It is easy to see that this is Markovian and the samples become less informative about the original $X$ as $t \to \infty$.

For the stochastic localization sampler, the posterior $Y_t | Y_{t+1}$ for adjacent $t, t + 1 \in \mathbf{I}$ is the conditional distribution of $T - t$-length sequences given a prefix of length $T - (t + 1)$, exactly the task of next-token prediction.

Now apply the forward-reverse experiment to a story of a cat. In this case, this means masking the last $\widehat{T}$ tokens of a sample and then resampling with the same model. If $\widehat{T}$ is small, the story will likely still mention a cat and resampling will yield a story about a cat. If $\widehat{T}$ is large, then the first appearance of cat may be truncated, so resampling could produce a story about a dog as well.

## 3. Characterization of critical windows

For expositional clarity, we will introduce our theory along with a toy setting to provide some useful intuition: consider an image diffusion model that outputs pictures of orange cats, brown cats, and dogs and a hypothetical critical window in which we transition from sampling from both cats and dogs to cats with certainty (Figure 2). In this case, the critical window simply refers to the dramatic increase in the probability of sampling a cat between time interval $[T_{\text{after}}^C, T_{\text{before}}^C]$. Note that the intuition heretofore will apply to all localization-based samplers with a forward and reverse process indexed by a time $t \in \mathbb{R}^+$.

Importantly, the critical window is situated between and in some sense defined by the two intervals which surround it: the left interval indicates that we have committed to only sampling from the distribution of cats indexed by $S_{\text{after}}$ and have a high probability of producing a cat as the final image, and the right interval, indexed by $S_{\text{before}}$, indicates we can sample images of cats or dogs. Thus critical windows simply represent the *speciation* from sampling from a larger sub-population of the distribution given by $S_{\text{before}}$ to a smaller sub-population given by $S_{\text{after}}$. In other words, we want to understand at what interval $t \in [B_1, B_2]$ is some *feature* (herein the feature of being a cat or dog) determined by the diffusion process and then at what interval $t \in [A_1, A_2]$ the more *specific version of the feature* (herein the feature

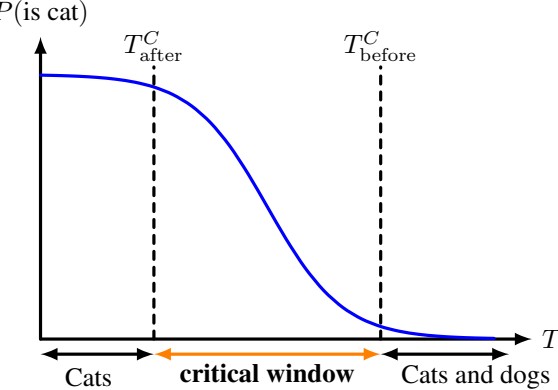

*Figure 2.* Illustration of a critical window for a cat feature with an image diffusion model.

of being a cat) is determined by the diffusion, yielding the critical window as the transition period $[A_2, B_1]$ in which we specialize to the more specific version of the feature. We can characterize both of these intervals $[A_1, A_2], [B_1, B_2]$ via the forward-reverse experiment.

More generally, let $S_{\text{init}} \subset \Theta$ denote some sub-mixture, corresponding to a sub-population of $p$ that possesses a certain property. Let $S_{\text{targ}} \supset S_{\text{init}}$ denote some sub-mixture containing $S_{\text{init}}$. For instance, $S_{\text{targ}}$ might correspond to all possible responses to the math question, including incorrect ones. We are interested in the following question: if we run the forward-reverse experiment for time $\widehat{T}$ starting from $p^{S_{\text{init}}}$, is there some range of times for which the resulting distribution is close to $p^{S_{\text{targ}}}$? That is, can we characterize the $\widehat{T}$ for which

$$\text{TV}(p^{S_{\text{init}}, \widehat{T}}, p^{S_{\text{targ}}})$$

is small?

Suppose one could prove that the range of $\widehat{T}$ for which this is the case is some interval $[T_0, T_1]$. This would mean that if the stochastic localization sampler runs for time $T$ and ends up at a sample from $p^{S_{\text{init}}}$, then from time $T - T_1$ to time $T - T_0$ of the generation process, *the sampler has not yet localized the features that distinguish $p^{S_{\text{init}}}$ from the larger sub-mixture $p^{S_{\text{targ}}}$*. However, the sampler has localized the features that distinguish $p^{S_{\text{targ}}}$ from $p^{\Theta - S_{\text{targ}}}$. When there is a shift from localizing the features $S_{\text{targ}}$ to the features $S_{\text{init}}$, we say there is a critical window. We now formally state and prove our main result.

### 3.1. Main result

We will parameterize the interval by the total variation distance between sub-mixtures $S_{\text{init}}$ and $S_{\text{targ}}$ along the for-

ward process. For an error parameter $0 < \epsilon < 1$, define

$$T_{\text{st}}^S(\epsilon) \in \sup\{t \in \mathbf{I} : \text{TV}(p_t^{S_{\text{targ}}}, p_t^{\Theta - S_{\text{targ}}}) \geq 1 - \epsilon^2\}$$
$$T_{\text{end}}^S(\epsilon) \in \inf\{t \in \mathbf{I} : \text{TV}(p_t^{S_{\text{init}}}, p_t^{S_{\text{targ}}}) \leq \epsilon\}.^4$$

When the value of $\epsilon$ is understood, we abbreviate the above with $T_{\text{st}}^S$ and $T_{\text{end}}^S$. Our main result is that in $\widehat{T} \in \mathbf{I} \cap [T_{\text{end}}^S, T_{\text{st}}^S]$, the distance $\text{TV}(p^{S_{\text{init}}, \widehat{T}}, p^{S_{\text{targ}}})$ is small:

**Theorem 3.1.** *Let* $S_{\text{init}} \subset S_{\text{targ}} \subset \Theta$ *and* $W = \frac{\sum_{\theta \in \Theta - S_{\text{targ}}} w_\theta}{\sum_{\theta \in S_{\text{targ}}} w_\theta}$. *For* $\epsilon > 0$, *if* $\widehat{T} \in \mathbf{I} \cap [T_{\text{end}}^S, T_{\text{st}}^S]$, *then* $\text{TV}(p^{S_{\text{init}}, \widehat{T}}, p^{S_{\text{targ}}}) \leq \epsilon \cdot \left(1 + \max\left(1, W\right) / \sqrt{2}\right)$.

Intuitively, $T_{\text{st}}^S$ represents the largest $t$ for which there is still separation between $S_{\text{targ}}$ and $\Theta - S_{\text{targ}}$, and $T_{\text{end}}^S$ represents the smallest $t$ for which samples from $S_{\text{init}}, S_{\text{targ}}$ are indistinguishable. Thus, running it for $\widehat{T} \in \mathbf{I} \cap [T_{\text{end}}^S, T_{\text{st}}^S]$ erases the differences between samples from $S_{\text{init}}$ and $S_{\text{targ}}$ but preserves the difference between $S_{\text{targ}}$ and $\Theta - S_{\text{targ}}$, yielding samples looking like $p^{S_{\text{targ}}}$.

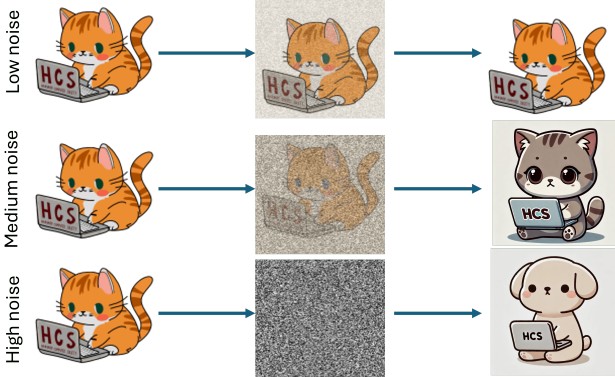

*Figure 3.* Intuition with the forward-reverse experiment with $S_{\text{init}} = \{\texttt{orange cat}\}$ and $S_{\text{targ}} = \{\texttt{orange cat}, \texttt{brown cat}\}$. In the low-noise regime, the forward-reverse experiment does not modify the image non-trivially, meaning we still sample from $S_{\text{init}}$; in the high-noise regime, the forward-reverse experiment destroys all information in the image and thus could yield a dog, lying outside of $S_{\text{targ}}$. Thus there is a "sweet spot" where the differences between orange and brown cats collapse but those between cats and dogs persist.

*A priori* it is not clear why $T_{\text{end}}^S$ should be smaller than $T_{\text{st}}^S$, and indeed in general it need not be and our bound would be vacuous. In Section 4 however, we show that in many natural settings for diffusion models and autoregressive models, the relation does hold.

*Remark* 3.2. Note a similar bound was shown in the context of diffusions by Li & Chen (2024) (see Theorem 7 therein). Our result is a strict improvement of that bound along several important axes. First, our results apply to all stochastic

localization samplers, not just diffusions. Secondly, Li & Chen (2024) needed to assume that the components of $p$ were strongly log-concave and that the score, i.e. gradient of the log-density, of $p_t$ was Lipschitz and moment-bounded for all $t$. Thirdly, their final bound includes a polynomial dependence on the moments of the score, which scale with the dimension $d$; in contrast, our final bound is independent of $d$.

With Theorem 3.1 in place, we are ready to formally define *critical windows*. These capture the moments where we transition from sampling from a sub-mixture to a subset of that sub-mixture.

**Definition 3.3.** Define $S_{\text{after}} \subset S_{\text{before}} \subset \Theta$. For $S_{\text{before}}$, we define $T_{\text{before}}^C = \inf\{t \in \mathbf{I} : \text{TV}(p_t^{S_{\text{after}}}, p_t^{S_{\text{before}}}) \leq \epsilon \text{ and } \text{TV}(p_t^{S_{\text{before}}}, p_t^{\Theta - S_{\text{before}}}) \geq 1 - \epsilon^2\}$ $(S_{\text{init}} \triangleq S_{\text{after}}; S_{\text{targ}} \triangleq S_{\text{before}})$. For $S_{\text{after}}$, consider $T_{\text{after}}^C = \sup\{t \in \mathbf{I} : \text{TV}(p_t^{S_{\text{after}}}, p_t^{\Theta - S_{\text{after}}}) \geq 1 - \epsilon^2\}$ $(S_{\text{init}}, S_{\text{targ}} \triangleq S_{\text{after}})$. A **critical window** is the interval $[T_{\text{after}}^C, T_{\text{before}}^C]$, where there is a transition from sampling from $S_{\text{before}}$ to the smaller subset $S_{\text{after}}$.

Finally, note that this definition does not directly explain why or when a critical window is sharp, which depends on the properties of the model of the data and forward process. In Section 4, we analytically compute critical windows in different toy settings and demonstrate that they are sharp in a natural sense. These examples also provide general intuition where critical windows are sharp, i.e. for autoregressive LLMs, critical windows may occur when a few tokens are very unlikely under one mode than other in Example 4.3. In general, we intuit that sharp critical windows occur when it only takes a few steps from the forward process to erase the differences between $S_{\text{before}}$ and $S_{\text{after}}$. This could happen if the data has a multi-scale hierarchical structure, where a feature is decided in a narrow intermediate band of the tree (Section 5).

We finally define some helpful notation for our proof. For nonempty $S \subset \Theta$ and $t \in \mathbf{I}$, we define $P^{\leftarrow}(\cdot | Y_t, S)$ to be the posterior of $X$ with the prior $X \sim p^S$. We similarly define $P_{t \to \Theta}^{\leftarrow}(\cdot | Y_t)$ and $P_{t \to \Theta}^{\leftarrow}(\cdot | Y_t, S)$ to be the posterior of $\theta$ conditioning on $Y_t$ with $X \sim p$ or $X \sim p^S$, respectively. When $S = \{i\}$, we exclude the braces.

### 3.2. Proof outline

Crucially, our proof relies in several places on the Markov property of stochastic localization samplers, together with the data processing inequality.

*Proof of Theorem 3.1.* By the triangle inequality, we can write $\text{TV}(p^{S_{\text{init}}, \widehat{T}}, p^{S_{\text{targ}}}) \leq \underbrace{\text{TV}(p^{S_{\text{init}}, \widehat{T}}, p^{S_{\text{targ}}, \widehat{T}})}_{(I)} + \underbrace{\text{TV}(p^{S_{\text{targ}}, \widehat{T}}, p^{S_{\text{targ}}})}_{(II)}.$ $p^{S_{\text{init}}, \widehat{T}}$

and $p^{S_{\text{targ}},\widehat{T}}$ are the laws of the posterior $P^{\leftarrow}(\cdot|\cdot)$ but applied to $Y_{\widehat{T}}$ with distributions $p_{\widehat{T}}^{S_{\text{init}}}$ and $p_{\widehat{T}}^{S_{\text{targ}}}$. Using the Markov property of localization-based samplers (Definition 2.2), we apply the data processing inequality and the definition of $T_{\text{end}}^S$ to bound (I) via $\text{TV}(p^{S_{\text{init}},\widehat{T}}, p^{S_{\text{targ}},\widehat{T}}) \leq \text{TV}(p_{T_{\text{end}}^S}^{S_{\text{init}}}, p_{T_{\text{end}}^S}^{S_{\text{targ}}}) \leq \epsilon$.

To bound (II), we apply the definition of TV and a coupling argument. By the law of total probability we can express $p^{S_{\text{targ}},\widehat{T}}(x) = \mathbb{E}[P^{\leftarrow}(x|Y_{\widehat{T}})]$ and $p^{S_{\text{targ}}}(x) = \mathbb{E}[P^{\leftarrow}(x|Y_{\widehat{T}}, S_{\text{targ}})]$ for $Y_{\widehat{T}} \sim p_{\widehat{T}}^{S_{\text{targ}}}$, as these observation processes have the same distribution at index $\widehat{T}$. Thus, $\text{TV}(p^{S_{\text{targ}},\widehat{T}}, p^{S_{\text{targ}}}) = \frac{1}{2}\int \left|p^{S_{\text{targ}},\widehat{T}}(x) - p^{S_{\text{targ}}}(x)\right| dx = \frac{1}{2}\int \left|\mathbb{E}[P^{\leftarrow}(x|Y_{\widehat{T}})] - \mathbb{E}[P^{\leftarrow}(x|Y_{\widehat{T}}, S_{\text{targ}})]\right| dx$. By Jensen's inequality and Fubini's theorem, we bring the expectation outside the integral, $\text{TV}(p^{S_{\text{targ}},\widehat{T}}, p^{S_{\text{targ}}}) \leq \frac{1}{2}\int \mathbb{E}\left[\left|P^{\leftarrow}(x|Y_{\widehat{T}}) - P^{\leftarrow}(x|Y_{\widehat{T}}, S_{\text{targ}})\right|\right] dx = \frac{1}{2}\mathbb{E}\left[\int \left|P^{\leftarrow}(x|Y_{\widehat{T}}) - P^{\leftarrow}(x|Y_{\widehat{T}}, S_{\text{targ}})\right| dx\right]$.

Combining Lemmas C.2 and C.3 (proved in Appendix C), we find $\text{TV}(p^{S_{\text{targ}},\widehat{T}}, p^{S_{\text{targ}}}) \leq \max(1, W)\,\mathbb{E}\left[\frac{p_{\widehat{T}}^{\Theta - S_{\text{targ}}}(Y_{\widehat{T}})}{p_{\widehat{T}}^{\Theta - S_{\text{targ}}}(Y_{\widehat{T}}) + p_{\widehat{T}}^{S_{\text{targ}}}(Y_{\widehat{T}})}\right]$. Then, finally applying Lemma 2.1, we are able to bound the total variation in terms of $\epsilon$, obtaining $\text{TV}(p^{S_{\text{targ}},\widehat{T}}, p^{S_{\text{targ}}}) \leq \frac{1}{2}\max(1, W)\sqrt{1 - \text{TV}^2(p_{\widehat{T}}^{\Theta - S_{\text{targ}}}, p_{\widehat{T}}^{S_{\text{targ}}})} \leq \frac{\sqrt{2}}{2}\max(1, W)\epsilon$. Combining our bounds on (I) and (II) achieves the desired result. $\square$

# 4. Examples of critical windows

In this section, we analytically compute $T_{\text{before}}^C, T_{\text{after}}^C$ for diverse stochastic localization samplers and models of data, including diffusion and autoregression processes. In these natural settings, the critical window is small in the sense of having a size which shrinks or does not depend on the dimension or context length. We shall also connect our framework to in-context learning and the all-or-nothing phenomenon. [5]

## 4.1. Diffusion

We first consider two examples of Gaussian Mixture Models and a diffusion model. We show that with two isotropic Gaussians, the critical window appears around a single point, $\ln\|\mu\|$, with width independent of the dimension.

*Example* 4.1. [Two Isotropic Gaussians] Let $\Theta = \{\pm 1\}$, $p^{+1} = \mathcal{N}(\mu, \text{Id})$, $p^{-1} = \mathcal{N}(-\mu, \text{Id})$. Then, we have a critical window transitioning from sampling from both components to the component $+1$ between $T_{\text{before}}^C = \ln\|\mu\| + \ln 2 + \ln 1/\epsilon$ and $T_{\text{after}}^C = \ln\|\mu\| - \ln\ln\frac{1}{2\epsilon^2}$. When

---

[5] Proofs are deferred to Appendix D.

$\widehat{T} \leq T_{\text{after}}^C$, then $\text{TV}(p^{+1,\widehat{T}}, p^{+1}) \lesssim \epsilon$. When $\widehat{T} \geq T_{\text{before}}^C$, $\text{TV}(p^{+1,\widehat{T}}, p) \lesssim \epsilon$.

For an isotropic Gaussian mixture model with randomly selected means, the critical window between sampling from one component to the entire mixture is also narrow. Note that we derive dimension-free widths in Example 4.2, an improvement over (Li & Chen, 2024) who had a $\ln\ln d$ dependence on dimension for isotropic Gaussians.

*Example* 4.2. [Random mean spherical Gaussians] We first sample $\mu_i \sim \mathcal{N}(0, \text{Id})$ for $i \in [K]$ i.i.d. and let $\Theta = \{\mathcal{N}(\mu_i, \text{Id})\}_{i \in [K]}$. We let $S_{\text{before}} = \Theta$ and $S_{\text{after}} = \{\mu_1\}$. Then, we can compute $T_{\text{before}}^C = \max_{j \in [K]} \ln\|\mu_i - \mu_j\| + \ln(1/\epsilon)$ and $T_{\text{after}}^C = \min_{j \in [K], i \neq j} \ln\|\mu_i - \mu_j\| - \frac{1}{2}\ln 8 \ln\frac{K}{\epsilon}$. Furthermore, with high probability over the selection of the means, $T_{\text{before}}^C - T_{\text{after}}^C = O_{K,\epsilon}(1)$ as $d \to \infty$.

Example D.5 of a discrete diffusion model, where $T_{\text{before}}^C - T_{\text{after}}^C \to 0$, is deferred to Appendix D.5.

## 4.2. Autoregression

We first present a theoretical model for important critical windows in LLMs, e.g., jailbreaks that occur over the first few tokens in the generation and the Yellowstone example (Anthropic, 2024; Qi et al., 2024).

*Example* 4.3. ["Critical Tokens" for Jailbreaks and Yellowstone (Qi et al., 2024; Anthropic, 2024)] Again consider an autoregressive language model, with $\mathcal{A}$ denoting the vocabulary, $p \in \mathcal{A}^T$, a forward process indexed by $\mathbf{I} = \{0, 1, 2, \ldots, T\}$, and $Y_t$ to be the first $T - t$ tokens of $X$. Let $\Theta = \{\theta_{\textbf{harmful}}, \theta_{\textbf{safe}}\}$ (or $\{\theta_{\textbf{Googling Yellowstone}}, \theta_{\textbf{coding}}\}$). We assume that these two modes do not differ until some $T - T' \in \mathbf{I}$. Between $T - T'$ and $T - T' - k$, the distributions become nearly disjoint, $P_{x \sim p_{T-T'-k}^{\theta_{\textbf{harmful}}}}\left(x \in \text{supp}(p_{T-T'-k}^{\theta_{\textbf{safe}}})\right) \leq \epsilon$. In the jailbreaking example, $T' = 0$ and they are disjoint because the first tokens generated in the safe mode is always some form of refusal. In the Yellowstone example, they are disjoint the first time the agent decides to Google Yellowstone pictures. Then, on component $\theta_{\textbf{harmful}}$ we have the critical window $T_{\text{before}}^C = T - T'$ and $T_{\text{after}}^C = T - T' - k$.

This prediction that the critical windows are narrow in these settings is verified by experiments from (Haize Labs, 2024b; Qi et al., 2024), which show that one consequently observes critical windows for jailbreaks. Moreover, notice that one can actually mitigate the effect of these critical windows by finetuning on examples of corrections to increase $P_{x \sim p_{T-T'-k}^{\theta_{\textbf{harmful}}}}\left(x \in \text{supp}(p_{T-T'-k}^{\theta_{\textbf{safe}}})\right)$. This explains the effectiveness of finetuning on corrections in (Qi et al., 2024).

*Remark* 4.4. The quantity that measures probability of mode-switching, $p^{\theta_{\textbf{harmful}}}/p$, suggests using a likelihood ra-

tio to distinguish between harmful and benign prompts. In App. F.2, we test a class of likelihood ratio methods that obtain recall 5-10× the false positive rate for 5 different types of jailbreaks (Table 2).

We defer critical window for a stylized model of solving a math problem as a random walk (Example D.6) and a critical window for an autoregressive model which expresses the outputs as emissions from a random walk of an underlying concept variable, akin to the model in (Arora et al., 2019), to Appendix D.2.

### 4.2.1. IN-CONTEXT LEARNING

Autoregressive critical windows can also be applied to describe in-context learning. In particular, we can capture the idea that with sufficiently many in-context examples, we learn the $\theta^* \in \Theta$ that generated the transitions for in-context examples, with a sample complexity in terms of $T_{\text{after}}^C$.

*Example* 4.5 (Informal, see Example D.16). Consider an in-context learning setup, where the context $[x_1, y_1, o, \ldots, x_{T+1}, y_{T+1}, o]$ consists of question-answer pairs $(x_i, y_i)$, delimiters $o$, and $x_i \to y_i$ sampled from $p^{\theta^*}$ for some $\theta^* \in \Theta$. In the forward-reverse experiment, we truncate it to $[x_1, y_1, o, \ldots, x_{T+1}]$, and then resample with $p$ to produce $[x_1, \ldots, x_{T+1}, \tilde{y}_{T+1}, o]$. The total variation between the sequences $[x_1, y_1, o, \ldots, x_{T+1}]$ and $[x_1, \ldots, x_{T+1}, \tilde{y}_{T+1}, o]$ can be viewed as the average-case error of the in-context learner and can be bounded within our critical windows framework. We have $T_{\text{after}}^C = 3T + 3 - O_\epsilon(1)$, with $O_\epsilon(1)$ independent of $T$ ($S_{\text{after}} \triangleq \{\theta^*\}$). Note that $T_{\text{after}}^C$ is the order of how many samples that can be erased so that we still are able to learn $\theta^* \in \Theta$.

One might ask if there is a $T_{\text{before}}^C$ for in-context learning, a threshold such that it is impossible to distinguish between $S_{\text{after}}, S_{\text{before}}$ with that many samples. In Appendix D.4, we will provide an example of a $T_{\text{before}}^C$ with the all-or-nothing phase transition from statistical inference.

## 5. Hierarchies in stochastic localization samplers

Thus far we have largely focused on a specific critical window, which would correspond to the formation of specific feature in the generated output. In some cases (Figure 8), there could be a sequence of critical windows, motivating a theory of hierarchical sampling within our critical windows framework.[6] This is naturally represented as a tree, where the vertices correspond to the cluster at some resolution of the forward process, the root is the entire data distribution, and the leaves are raw components of the distribution. A path from the root to a leaf captures the progressive refine-

---

[6]Details in Appendix E.

ment of the original distribution $p$ into increasingly specific components. To formalize this, we introduce the concept of an $\epsilon$-mixture tree, which decomposes $p$ into a hierarchical structure.

**Definition 5.1** (Informal, see Definition E.1). For an error term $\epsilon > 0$ and mixture model $p$, an $\epsilon$-**mixture tree** is a tuple $(T, \{P^{\to}(\cdot|\cdot)\}, \mathbf{I}, \Theta, \{p^\theta\}_{\theta \in \Theta}, \text{Subset}, \text{NoiseAmount})$. $T = (V, E)$ is a tree associated with a function $\text{Subset}: V \to 2^\Theta \setminus \{\emptyset\}$, which maps vertices to subsets of $\Theta$ and $\text{NoiseAmount}: V \to \mathbb{R}^{\geq 0}$, which characterizes the noises levels for different aggregations in the mixture tree. If $u$ is a parent of $v$, $\text{Subset}(v) \subset \text{Subset}(u)$ for $u \in V$; $\text{NoiseAmount}(u)$ is defined such that all $p^\theta_{\text{NoiseAmount}(u)}$ for $\theta \in \text{Subset}(u)$ overlap greatly and for $p^{\text{Subset}(u)}_{\text{NoiseAmount}(u)}, p^{\Theta - \text{Subset}(u)}_{\text{NoiseAmount}(u)}$ have negligible overlap.

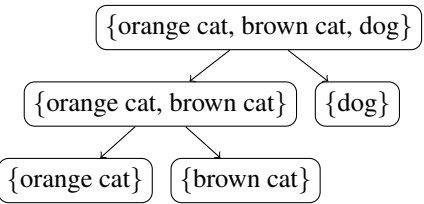

*Figure 4.* Example of an $\epsilon$-mixture tree for a diffusion model that produces images of orange cats, brown cats, and dogs.

We emphasize that this framework is highly general, solely defined with the *initial distribution $p$* and *the forward process*. It strictly expands the definition in (Li & Chen, 2024), which focused on hierarchies of isotropic Gaussians, to all localization-based samplers and mixture models. We can also relate it to the sequences of critical windows we observe in Figure 8, capturing the idea that each critical window represents the refinement into smaller subpopulations of $p$.

**Corollary 5.2.** *Consider an $\epsilon$-mixture tree. For $\theta_i \in \Theta$, consider the path $u_1, u_2, u_3, \ldots, u_{H'} \in V$ where $u_1$ is the leaf node with $\theta_i \in \text{Subset}(u_1)$ and $u_{H'}$ is the root. There is a sequence of times $T_1 < T_2 < \cdots < T_H$ with $\text{TV}(p^{\{i\}, T_\ell}, p^{\text{Subset}(u_\ell)}) \lesssim_w \epsilon$.*

We first observe that the hierarchy of two samplers with the same forward process are identical if the samplers agree on sub-populations. Assume we have $\{p^\theta\}_{\theta \in \Theta}$ (e.g. the true distribution) and $\{q^\theta\}_{\theta \in \Theta}$ (e.g. a generative model), where $q^\theta \approx p^\theta$ across all $\theta \in \Theta$ with the same $\{w_\theta\}_{\theta \in \Theta}$.

**Corollary 5.3.** *Consider an $\epsilon$-mixture tree $(T, \{P^{\to}(\cdot|\cdot)\}, \mathbf{I}, \Theta, \{p^\theta\}_{\theta \in \Theta}, \text{Subset}, \text{NoiseAmount})$. Suppose we have another distribution $\{q^\theta\}_{\theta \in \Theta}$ such that $\text{TV}(p^\theta, q^\theta) \leq \delta/2$ for all $\theta \in \Theta$. Then we have $\epsilon + \sqrt{\delta}$-mixture tree given by $(T, \{P^{\to}(\cdot|\cdot)\}, \mathbf{I}, \Theta, \{q^\theta\}_{\theta \in \Theta}, \text{Subset}, \text{NoiseAmount})$.*

This similarity does not hold generally. We can define arbitrary hierarchies by choosing the right forward process.

*Example* 5.4. Consider a set of alphabets $\{\mathcal{A}_i\}_{i=1}^d$ and define $\Theta = \{(a_i)_{i=1}^d : \forall i \in [d], a_i \in \mathcal{A}_i\}$ and $p^{\theta_i} = \delta_{\theta_i}$. Let $\mathbf{I} = [0, 1, 2, \ldots, d]$. and for any permutation $i_1, i_2, \ldots, i_d$ of $[d]$, define a forward process such that at $t \in \mathbf{I}$, we mask all $i_d, i_{d-1}, \ldots, i_{d-t}$. This constructs a hierarchy where the values for $i_1, i_2, \ldots, i_d$ are decided in that order.

Finally, we note that hierarchies of diffusions are in general shallower than for autoregressive models. The hierarchy for a mixture of Gaussians has depth $O(1)$ (Example 4.2), as the forward process simultaneously contracts all distances with a similar dependence on $d$. For autoregressive models, depth can scale linearly with the context length (Example 5.4). While they refer to different modeling approaches, it is worthwhile to highlight this vast difference in hierarchy depth can vary with the dimension. We speculate that this could mean autoregressive models can learn more complex feature hierarchies than diffusions.

## 6. Experiments

As many authors[7] have empirically studied critical windows for diffusion and verified different theoretical predictions with experiments for this model, we focus on experiments on critical windows for LLMs. In Appendix G, we validate the predictions of our theory on outputs with a hierarchical structure, showing strong agreement with Section 5.

Then we identify critical windows for LLAMA-3.1-8B-Instruct, Phi-3-7B-Instruct, and Qwen-2.5-7B-Instruct in the correctness of real-world math and reasoning benchmarks.[8] In our forward-reverse experiments to identify critical windows, we take the original generation, truncate a fraction of tokens, and check if resampling yields the same answer. We say that a particular generation has a critical window if there is a $> 0.5$ jump in probability of obtaining the same answer in consecutive truncation fractions.

We find that critical windows appear prominently across all benchmarks (Figure 10 to 12); for MATH, they occur in $42.2\%, 16.6\%, 30.2\%$ of generations from LLAMA-3.1-8B-Instruct, Qwen-2.5-7B-Instruct, and Phi-3-7B-Instruct. These jumps seem to occur during important steps of reasoning, as certain parts of the solution may be more significant to the formation of the final answer. For example, in Figure 5, the model restates the correct formula to apply to the problem within the critical window. For other generations, it could be that no particular part of the text is especially im-

---

[7](Ho et al., 2020; Meng et al., 2022; Choi et al., 2022; Raya & Ambrogioni, 2023; Georgiev et al., 2023)

[8]See Appendix H for full results across models and datasets and a discussion on the effect of temperature on critical windows.

*Table 1.* Differences between Accuracy (Acc) without versus with critical windows and frequency of critical windows (CW) when the original generation is wrong versus right.

| Dataset | $\Delta$**Acc** | $\Delta$**CW** |
|---|---|---|
| AQUA-RAT | 0.42 | 0.20 |
| ARC Challenge | 0.53 | 0.22 |
| ARC Easy | 0.73 | 0.26 |
| LogiQA | 0.15 | 0.11 |
| MATH | 0.41 | 0.33 |
| MMLU | 0.45 | 0.24 |
| TruthfulQA | 0.44 | 0.20 |

**MATH (Phi-3-7B-Instruct)**

**Q**: If the first term of an arithmetic sequence equals the third minus the second, and the fourth term is 8, what is the first term? **A**: 1. We know that the arithmetic sequence has a common difference, which we will denote as "$d$". 2. Let's denote the terms in the sequence as $T_1$, $T_2$, $T_3$, and $T_4$ for the first, second, third, and fourth terms, respectively. 3. We're given the information that $T_1 = T_3 - T_2$. Using the general formula for an arithmetic sequence, we can write this **as $T_1 = T_2 + 2d - T_2$, which simplifies to $T_1 = 2d$. 4. We're also given that $T_4 = 8$. The general formula for an arithmetic sequence tells us that $T_4 = T_1 + 3d$. So we can write** $8 = T_1 + 3d$. 5. Now we have a system of two equations: 1) $T_1 = 2d$ 2) $8 = T_1 + 3d$ 6. We can substitute equation (1) into equation (2), which gives us $8 = 2d + 3d = 5d$. 7. Solving for $d$, we get $d = 8/5$. 8. Now that we know d, we can find $T_1$. Using the equation from step 1, we get $T_1 = 2d = 2 * (8/5) = 16/5$. Therefore, the first term is $T_1 = 16/5$.

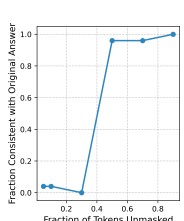

*Figure 5.* Example of critical window (bolded).

portant for the final answer, resulting in no critical window. In fact, we find that generations with critical windows are significantly *less accurate* than generations without critical windows across all datasets and models. For LLAMA-3.1-8B-Instruct, critical windows result in up to **73% lower accuracy**, and incorrect responses contain **11%-33% more critical windows** (Table 1).

## 7. Discussion

In this work, we developed a simple yet general theory for critical windows for stochastic localization samplers like diffusion and autoregressive models. Already, practitioners have applied critical windows to make LLMs safer (Qi et al., 2024) and reason better (Abdin et al., 2024; Lin et al., 2024). Our theory significantly streamlines our understanding of critical windows and provides concrete insights for practitioners. While we instantiate our theory in many settings (Section 4), an important future direction is connect it to statistical physics predictions for other models of data, e.g., the random hierarchy model (Sclocchi et al., 2024). Another interesting direction would be to explore why critical windows exist in different settings in practice.

## Acknowledgments

ML would like to thank Cynthia Dwork for regular meetings and kind mentorship which were of great help throughout this research project, Cem Anil, Michael Li, and Eric Zelikman for insightful conversations which inspired some of the experiments, and Max Simchowitz, Michael Albergo, Seth Neel, and Sahil Kuchlous for thoughtful discussions regarding the framing of this work.

## Impact statement

While this paper is largely theoretical in nature, it does describe a theory for jailbreaks which could impact model safety in the future. We hope the insights in this manuscript about jailbreaks lead to better alignment strategies and training methods.

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

# A. Additional related work

**Theory of in-context learning for language models.** With respect to theory for language models, our results are most closely related to the Bayesian framework for in-context learning (Xie et al., 2022; Akyürek et al., 2023; Garg et al., 2023; Zhang et al., 2023; Arora et al., 2024). For example, (Xie et al., 2022) also considered a mixture model of topics and showed that language models can learn the underlying class despite in-context learning and training distribution mismatch. We view this manuscript as connecting the Bayesian framework for in-context learning to other empirical phenomena observed in language models and diffusion models and the all-or-nothing phenomenon.

**Chain of thought.** (Lin et al., 2024; Abdin et al., 2024) also observed that the presence of critical windows in the chain of thought of math and reasoning tasks and their significance in leading the model to incorrect outputs, concurrent with our results in Figure 5. They then used them to provide rewards or data for a preference optimization algorithm to improve reasoning performance. (Lin et al., 2024) called them *critical tokens* and utilized a contrastive estimation algorithm to identify critical windows and provide token-level rewards. The Phi-4 Technical report called them *pivotal tokens*, developed a binary-search based algorithm to identify the location of critical windows, and used them to produce contrasting pairs for preference optimization (Abdin et al., 2024). Using our broad theoretical perspective, we provide new insight into critical windows of these kinds and view our work as corroborating and extending these empirical works.

**Jailbreaks.** Existing work on jailbreaks has studied the appearance of critical windows in the first few generated tokens (Qi et al., 2024; Zhang & Wu, 2024; He et al., 2024; Lin et al., 2023). Our theory provides a simple explanation for when jailbreaks occur: when the unaligned component assigns a much higher probability to the current text than the aligned component, then the model is jailbroken. This generalizes the explanation from (Qi et al., 2024) (see Example 4.3 for our particular formalism of their insights). It also explains the success of perplexity-based monitors for jailbreaks (Alon & Kamfonas, 2023), which monitor for a low probability of the context and generation. We view our work as providing a rigorous mathematical framework for jailbreaks, as well as highlighting the important role off-distribution contexts play in eliciting harmful behaviors; we also develop a novel jailbreak from our framework (Section F.2 ) similar to the adversarial contrast decoding method proposed by (Zhao et al., 2024), which also uses a likelihood ratio between an unaligned and an aligned model. However, we use a jailbroken and non-jailbroken pair of models instead of two versions of the model with different prompts.

# B. Examples of stochastic localization

In this section, we present several kinds of generative models within the stochastic localization framework and their forward and reverse processes.

*Example* B.1 (Continuous Diffusion Models (Li & Chen, 2024)). For continuous diffusion models, the forward process progressively degrades samples $X \sim p$ into pure Gaussian noise through scaling and convolution with Gaussian noise. It is the Ornstein-Uhlenbeck process, a stochastic process $(X_t)_{t \geq 0}$ given by the following stochastic differential equation (SDE),

$$\mathrm{d}X_t = -X_t \, \mathrm{d}t + \sqrt{2}\mathrm{d}B_t\,, \qquad X_0 \sim p\,,$$

where $(B_t)_{t \geq 0}$ is a standard Brownian motion. Let $q_t \triangleq \mathrm{law}(X_t)$ for $t \geq 0$, and observe that as $t \to \infty$, $q_t$ converges exponentially quickly to the standard Gaussian distribution $\mathcal{N}(0, \mathrm{Id})$. Assume we end the forward process at time $T \geq 0$. For the reverse process $(X_t^{\leftarrow})_{t \in [0,T]}$, we employ the *reversal* of the Ornstein-Uhlenbeck SDE, given by

$$\mathrm{d}X_t^{\leftarrow} = \{X_t^{\leftarrow} + 2\nabla \ln q_{T-t}(X_t^{\leftarrow})\} \, \mathrm{d}t + \sqrt{2} \, \mathrm{d}B_t\,, \qquad X_T^{\leftarrow} \sim q_T$$

where here $(B_t)_{t \geq 0}$ is also a Brownian motion. Defining $(Y_t)_{t \in \mathbf{I}} = (X_t)_{t \in \mathbf{I}}$, we see that the forward process satisfies the Markov property in Definition 2.2, and the information from the original sample $X_0$ is degraded by more steps in the SDE. Furthermore, the reverse SDE with parameterized by the score function $\nabla \ln q_{T-t}(X_t^{\leftarrow})$ can be viewed as successively sampling from the posteriors via Tweedie's formula.

*Example* B.2 (Discrete Diffusion Models (Lou et al., 2024)). Consider a set $\mathcal{A}$ denoting the vocabulary and let $p \in \mathcal{A}^T$, and consider a forward process with index set $\mathbf{I} = [0, K] \cup \{\infty\}$, $Y_0 = X$, and $Y_t \in \mathcal{A}^T$ defined in the limit as follows,

$$p(Y_{t+\Delta t} = a \mid Y_t = b) = \delta_{ab} + Q_t(b, a)\Delta t + O(\Delta t^2),$$

where $Q_t \in \mathbb{R}^{n \times n}$ are diffusion matrices with nonnegative non-diagonal entries and columns which sum to $0$. $(Y_t)_{t \in \mathbf{I}}$ is also a Markov chain and as $t \to \infty$, $Y_t$ is degraded until it is eventually uninformative about the original sample $Y_0$.

*Example* B.3 (Autoregressive Language Models (**?**)). Consider a set $\mathcal{A}$ denoting the vocabulary and let $p \in \mathcal{A}^T$, and consider a forward process with index set $\mathbf{I} = \{0, 1, 2, \ldots, T\}$, $Y_0 = X$, and $Y_t \in \mathcal{A}^{T-t}$. For $t \in \mathbf{I}$, we let $Y_t$ equal the last first $T - t$ tokens of $X$. Clearly this is a Markov Chain, and the reverse process is equivalent to next-token prediction.

*Remark* B.4. We note if the reverse process is deterministic, e.g. ODE-based diffusion models or language models at temperature $0$, then there is no notion of a critical window under our framework. The initial position at the start of sampling completely characterizes the final image. For example, given a fixed piece of text and language model, truncating it anywhere in the model's response and resampling at temperature $0$ will yield the original piece of text.

## C. Deferred details from Section 3

*Remark* C.1 (Technicality in defining $T_{\text{st}}^S, T_{\text{end}}^S$). For general stochastic localization schemes, we can only ask that $T_{\text{st}}^S(\epsilon) \in \{r \in \mathbf{I} : \text{TV}(p_t^{S_{\text{targ}}}, p_t^{\Theta - S_{\text{targ}}}) \geq 1 - \epsilon^2 \}$ and $T_{\text{end}}^S(\epsilon) \in \{t \in \mathbf{I} : \text{TV}(p_t^{S_{\text{init}}}, p_t^{S_{\text{targ}}}) \leq \epsilon\}$ instead of $\sup, \inf$ like (Li & Chen, 2024), because the sets $\{t \in \mathbf{I} : \text{TV}(p_t^{S_{\text{targ}}}, p_t^{\Theta - S_{\text{targ}}}) \geq 1 - \epsilon^2 \}, \{t \in \mathbf{I} : \text{TV}(p_t^{S_{\text{init}}}, p_t^{S_{\text{targ}}}) \leq \epsilon\}$ may not be closed for observation processes which are discontinuous. For autoregressive language models and continuous diffusion, the observation process is continuous, so we will elide these technicalities.

For Theorem 3.1, we employ the following two helper lemmas.

**Lemma C.2.** *By applying the law of total probability and Bayes' rule, we can show for $Y_{\widehat{T}} \in \text{supp}(p_{\widehat{T}}^{S_{\text{targ}}})$,*

$$\int \left| P^{\leftarrow}(x|Y_{\widehat{T}}) - P^{\leftarrow}(x|Y_{\widehat{T}}, S_{\text{targ}}) \right| dx \leq 2 \sum_{\theta \in \Theta - S_{\text{targ}}} P_{t \to \Theta}^{\leftarrow}(\theta|Y_{\widehat{T}}).$$

*Proof.* We can rewrite $P^{\leftarrow}(x|Y_{\widehat{T}}), P^{\leftarrow}(x|Y_{\widehat{T}}, S_{\text{targ}})$ using the law of total probability and Bayes' rule.

$$P^{\leftarrow}(x|Y_{\widehat{T}}) = \sum_{\theta \in \Theta} P_{t \to \Theta}^{\leftarrow}(\theta|Y_{\widehat{T}}) P^{\leftarrow}(x|Y_{\widehat{T}}, \theta)$$

$$P^{\leftarrow}(x|Y_{\widehat{T}}, S_{\text{targ}}) = \sum_{\theta \in S_{\text{targ}}} P_{t \to \Theta}^{\leftarrow}(\theta|Y_{\widehat{T}}, S_{\text{targ}}) P^{\leftarrow}(x|Y_{\widehat{T}}, \theta) = \frac{\sum_{\theta \in S_{\text{targ}}} P_{t \to \Theta}^{\leftarrow}(\theta|Y_{\widehat{T}}) P^{\leftarrow}(x|Y_{\widehat{T}}, \theta)}{\sum_{\theta \in S_{\text{targ}}} P_{t \to \Theta}^{\leftarrow}(\theta|Y_{\widehat{T}})}.$$

Note that the second equality on the second line follows from the fact that for all $\theta \in S_{\text{targ}}$, the posteriors $P_{t \to \Theta}^{\leftarrow}(\cdot|Y_t) \propto P_{t \to \Theta}^{\leftarrow}(\cdot|Y_t, S_{\text{targ}})$ by the same normalization constant. Therefore the difference can be written as

$$\int \left| P^{\leftarrow}(x|Y_{\widehat{T}}) - P^{\leftarrow}(x|Y_{\widehat{T}}, S_{\text{targ}}) \right| dx$$

$$= \int \left| \sum_{\theta \in \Theta} P_{t \to \Theta}^{\leftarrow}(\theta|Y_{\widehat{T}}) P^{\leftarrow}(x|Y_{\widehat{T}}, \theta) - \frac{\sum_{\theta \in S_{\text{targ}}} P_{t \to \Theta}^{\leftarrow}(\theta|Y_{\widehat{T}}) P^{\leftarrow}(x|Y_{\widehat{T}}, \theta)}{\sum_{\theta \in S_{\text{targ}}} P_{t \to \Theta}^{\leftarrow}(\theta|Y_{\widehat{T}})} \right| dx$$

$$= \int \left| \left( 1 - \frac{1}{\sum_{\theta \in S_{\text{targ}}} P_{t \to \Theta}^{\leftarrow}(\theta|Y_{\widehat{T}})} \right) \sum_{\theta \in S_{\text{targ}}} P_{t \to \Theta}^{\leftarrow}(\theta|Y_{\widehat{T}}) P^{\leftarrow}(x|Y_{\widehat{T}}, \theta) + \sum_{\theta \in \Theta - S_{\text{targ}}} P_{t \to \Theta}^{\leftarrow}(\theta|Y_{\widehat{T}}) P^{\leftarrow}(x|Y_{\widehat{T}}, \theta) \right| dx.$$

If $\sum_{\theta \in \Theta - S_{\text{targ}}} P_{t \to \Theta}^{\leftarrow}(\theta|Y_{\widehat{T}}) = 0$, then the above is equal to $0$ and we are done. If it is non-zero, we can factor out $\sum_{\theta \in \Theta - S_{\text{targ}}} P_{t \to \Theta}^{\leftarrow}(\theta|Y_{\widehat{T}})$ term, which allows us to write everything in terms of posteriors with respect to $\Theta - S_{\text{targ}}$ and $S_{\text{targ}}$,

$$\int \left| \left( 1 - \frac{1}{\sum_{\theta \in S_{\text{targ}}} P_{t \to \Theta}^{\leftarrow}(\theta|Y_{\widehat{T}})} \right) \sum_{\theta \in S_{\text{targ}}} P_{t \to \Theta}^{\leftarrow}(\theta|Y_{\widehat{T}}) P^{\leftarrow}(x|Y_{\widehat{T}}, \theta) + \sum_{\theta \in \Theta - S_{\text{targ}}} P_{t \to \Theta}^{\leftarrow}(\theta|Y_{\widehat{T}}) P^{\leftarrow}(x|Y_{\widehat{T}}, \theta) \right| dx$$

$$= \sum_{\theta \in \Theta - S_{\text{targ}}} P_{t \to \Theta}^{\leftarrow}(\theta|Y_{\widehat{T}}) \int \left| P^{\leftarrow}(x|Y_{\widehat{T}}, S_{\text{targ}}) - P^{\leftarrow}(x|Y_{\widehat{T}}, \Theta - S_{\text{targ}}) \right| dx$$

Employing the trivial observation that

$$\int \left| P^{\leftarrow}(x|Y_{\widehat{T}}, S_{\text{targ}}) - P^{\leftarrow}(x|Y_{\widehat{T}}, \Theta - S_{\text{targ}}) \right| dx$$

$$\leq \int P^{\leftarrow}(x|Y_{\widehat{T}}, S_{\text{targ}}) + P^{\leftarrow}(x|Y_{\widehat{T}}, \Theta - S_{\text{targ}}) dx \leq 2,$$

we have

$$\int \left| P^{\leftarrow}(x|Y_{\widehat{T}}) - P^{\leftarrow}(x|Y_{\widehat{T}}, S_{\text{targ}}) \right| dx \leq 2 \sum_{\theta \in \Theta - S_{\text{targ}}} P^{\leftarrow}_{t \to \Theta}(\theta|Y_{\widehat{T}}).$$

$\square$

**Lemma C.3.** *By Bayes's rule, we can derive for $Y_{\widehat{T}} \in \text{supp}(p_{\widehat{T}})$,*

$$\sum_{\theta \in \Theta - S_{\text{targ}}} P^{\leftarrow}_{t \to \Theta}(\theta|Y_{\widehat{T}}) \leq \max(1, W) \frac{p_{\widehat{T}}^{\Theta - S_{\text{targ}}}(Y_{\widehat{T}})}{p_{\widehat{T}}^{\Theta - S_{\text{targ}}}(Y_{\widehat{T}}) + p_{\widehat{T}}^{S_{\text{targ}}}(Y_{\widehat{T}})}$$

*Proof.* We obtain through Bayes' rule,

$$\sum_{\theta \in \Theta - S_{\text{targ}}} P^{\leftarrow}_{t \to \Theta}(\theta|Y_{\widehat{T}}) = \frac{\sum_{\theta \in \Theta - S_{\text{targ}}} w_\theta p_{\widehat{T}}^\theta(Y_{\widehat{T}})}{\sum_{\theta \in \Theta} w_\theta p_{\widehat{T}}^\theta(Y_{\widehat{T}})}.$$

We divide by the same normalizing constant $\sum_{\theta \in \Theta - S_{\text{targ}}} w_\theta$ to obtain

$$\frac{\sum_{\theta \in \Theta - S_{\text{targ}}} w_\theta p_{\widehat{T}}^\theta(Y_{\widehat{T}})}{\sum_{\theta \in \Theta} w_\theta p_{\widehat{T}}^\theta(Y_{\widehat{T}})} = \frac{\frac{\sum_{\theta \in \Theta - S_{\text{targ}}} w_\theta p_{\widehat{T}}^\theta(Y_{\widehat{T}})}{\sum_{\theta \in \Theta - S_{\text{targ}}} w_\theta}}{\frac{\sum_{\theta \in \Theta - S_{\text{targ}}} w_\theta p_{\widehat{T}}^\theta(Y_{\widehat{T}})}{\sum_{\theta \in \Theta - S_{\text{targ}}} w_\theta} + \frac{\sum_{\theta \in S_{\text{targ}}} w_\theta p_{\widehat{T}}^\theta(Y_{\widehat{T}})}{\sum_{\theta \in S_{\text{targ}}} w_\theta} \cdot \frac{\sum_{\theta \in S_{\text{targ}}} w_\theta}{\sum_{\theta \in \Theta - S_{\text{targ}}} w_\theta}}$$

$$\leq \max\left(1, \frac{\sum_{\theta \in \Theta - S_{\text{targ}}} w_\theta}{\sum_{\theta \in S_{\text{targ}}} w_\theta}\right) \frac{p_{\widehat{T}}^{\Theta - S_{\text{targ}}}(Y_{\widehat{T}})}{p_{\widehat{T}}^{\Theta - S_{\text{targ}}}(Y_{\widehat{T}}) + p_{\widehat{T}}^{S_{\text{targ}}}(Y_{\widehat{T}})}.$$

$\square$

# D. Omitted details from Section 4

## D.1. Diffusions

Here we use an alternative $f$-divergence to characterize the critical windows, the squared *Hellinger distance*, defined as $\text{H}^2(P, Q) \triangleq \int (\sqrt{dP} - \sqrt{dQ})^2 d\mu$, because there are explicit computations for the Hellinger distance for mixtures of Gaussians. We similarly exploit the following ratio inequality akin to Lemma 2.1,

**Lemma D.1.** *For probability measures $P, Q$,*

$$\mathbb{E}_{x \sim P}\left[\frac{dQ}{dP + dQ}\right] \leq \frac{1}{2}\left(1 - \frac{1}{2}\text{H}^2(P, Q)\right).$$

We apply the following well-known formula for the Hellinger distance between two Gaussians.

**Lemma D.2.** *We have*

$$1 - \frac{1}{2}H^2(\mathcal{N}(\mu_P, \Sigma_P), \mathcal{N}(\mu_Q, \Sigma_Q)) = \frac{|\Sigma_P|^{1/4}|\Sigma_Q|^{1/4}}{\left|\frac{\Sigma_P + \Sigma_Q}{2}\right|^{1/2}} \exp\left\{-\frac{1}{8}(\mu_P - \mu_Q)^\top \left[\frac{\Sigma_P + \Sigma_Q}{2}\right]^{-1} (\mu_P - \mu_Q)\right\}.$$

*Example* D.3. [Two Isotropic Gaussians] Let $\Theta = \{\pm 1\}$, $p^{+1} = \mathcal{N}(\mu, \mathrm{Id})$, $p^{-1} = \mathcal{N}(-\mu, \mathrm{Id})$. Then, we have a critical window transitioning from sampling from both components to the component $+1$ between $T_{\text{before}}^C = \ln \|\mu\| + \ln 2 + \ln 1/\epsilon$ and $T_{\text{after}}^C = \ln \|\mu\| - \ln \ln \frac{1}{2\epsilon^2}$. When $\widehat{T} \le T_{\text{after}}^C$, then $\mathrm{TV}(p^{+1,\widehat{T}}, p^{+1}) \lesssim \epsilon$. When $\widehat{T} \ge T_{\text{before}}^C$, $\mathrm{TV}(p^{+1,\widehat{T}}, p) \lesssim \epsilon$.

*Proof.* The proof for $T_{\text{before}}^C$, a simple application of Pinsker's inequality, can be found in Appendix B.1 of (Li & Chen, 2024). Directly applying the new Master Theorem 3.1 to $T_{\text{after}}^C$, we need only show that

$$\frac{\sqrt{2}}{2} \left( 1 - \frac{1}{2} \mathrm{H}^2(p_{T_{\text{after}}^C}^{+1}, p_{T_{\text{after}}^C}^{-1}) \right) \le \frac{\sqrt{2}}{2} \exp\left(-\frac{1}{2} \|\mu\|^2 e^{-2t}\right) \le \epsilon.$$

$\square$

*Example* D.4. [Random mean spherical Gaussians] We first sample $\mu_i \sim \mathcal{N}(0, \mathrm{Id})$ for $i \in [K]$ i.i.d. and let $\Theta = \{\mathcal{N}(\mu_i, \mathrm{Id})\}_{i \in [K]}$. We let $S_{\text{before}} = \Theta$ and $S_{\text{after}} = \{\mu_1\}$. Then, we can compute $T_{\text{before}}^C = \max_{j \in [K]} \ln \|\mu_i - \mu_j\| + \ln(1/\epsilon)$ and $T_{\text{after}}^C = \min_{j \in [K], i \ne j} \ln \|\mu_i - \mu_j\| - \frac{1}{2} \ln 8 \ln \frac{K}{\epsilon}$. Furthermore, with high probability over the selection of the means, $T_{\text{before}}^C - T_{\text{after}}^C = O_{K,\epsilon}(1)$ as $d \to \infty$.

*Proof.* The proof for $T_{\text{before}}^C$ can be found in Section 5.2 of (Li & Chen, 2024). We need to slightly modify the proof of Theorem 3.1 so that we can write the desired bound for $T_{\text{after}}^C$ in terms of the Hellinger distance of individual components. We use the same notation. By convexity, we can bound

$$\mathbb{E}_{Y_{\widehat{T}} \sim p_{\widehat{T}}^{S_{\text{targ}}}} \left[ \frac{\sum_{\theta \in \Theta - S_{\text{targ}}} w_\theta p_{\widehat{T}}^\theta(Y_{\widehat{T}})}{\sum_{\theta \in \Theta} w_\theta p_{\widehat{T}}^\theta(Y_{\widehat{T}})} \right] \le \sum_{\theta \in \Theta - S_{\text{targ}}} w_\theta \sum_{\phi \in S_{\text{targ}}} w_\phi \mathbb{E}_{Y_{\widehat{T}} \sim p_{\widehat{T}}^\phi} \left[ \frac{p_{\widehat{T}}^\theta(Y_{\widehat{T}})}{w_\theta p_{\widehat{T}}^\theta(Y_{\widehat{T}}) + w_\phi p_{\widehat{T}}^\phi(Y_{\widehat{T}})} \right]$$

$$\le K \max_{\theta \in \Theta - S_{\text{targ}}, \phi \in S_{\text{targ}}} \left( 1 - \frac{1}{2} \mathrm{H}^2(p_{\widehat{T}}^\theta, p_{\widehat{T}}^\phi) \right) \le \epsilon,$$

when $\widehat{T} \le T_{\text{after}}^C$. To conclude the second part of the theorem, observe that by concentration of measure (e.g., Theorem 3.1.1 from (Vershynin)) and a union bound, there exists a constant $T$ independent of $d$ such that $\|\mu_j\| \in [\sqrt{d} - T, \sqrt{d} + T]$ for all $j \in [K]$ with high probability. Furthermore, by known Gaussian Suprema inequalities, we can also assume that there exists a constant $T'$ independent of $d$ such that $|\langle \mu_i, \mu_j \rangle| \le T' \|\mu_i\|$ (Lemma 5.1 from (van Handel, 2016)). Thus, we can conclude that

$$\max_{j \in [K]} \|\mu_i - \mu_j\|^2 \le 2d + 4T\sqrt{d} + 2T^2 + 2T'(\sqrt{d} + T) = O(d).$$

$$\max_{j \in [K]} \|\mu_i - \mu_j\|^2 \ge 2d - 4T\sqrt{d} + 2T^2 - 2T'(\sqrt{d} + T) = \Omega(d).$$

The difference in $\log$ scale is thus constant,

$$\frac{1}{2} \left( \ln \max_{j \in [K]} \|\mu_i - \mu_j\|^2 - \ln \min_{j \in [K], i \ne j} \|\mu_i - \mu_j\|^2 \right) = O_{K,\epsilon}(1)$$

$\square$

*Example* D.5. [Two Dirac delta functions with a random masking procedure] Let $p \in \{\pm 1\}^T$, and consider a forward process with index set $\mathbf{I} = [0, 1]$, $Y_0 = X$, and $Y_t \in \{\pm 1, [\text{MASKED}]\}^T$. For $t \in \mathbf{I}$, we let all the value at index $i \in [T]$ be set to $[\text{MASKED}]$ with probability $t$ independently. For a mixture of two Dirac delta functions, we can express the critical window in terms of the *Hamming distance* between the corresponding strings. Let $\Theta = \{\theta_{\pm 1}\}$, $\ell_{\pm 1} \in \{\pm 1\}^T$, $p^{\theta_{\pm 1}} \sim \delta_{\ell_{\pm 1}}$, $w_{\pm 1} = \frac{1}{2}$. Then, on component 1 we have the critical window

$$T_{\text{before}}^C = \exp\left[ \frac{\ln(1 - \epsilon)}{d_H(\delta_{\ell_1}, \delta_{\ell_{-1}})} \right], T_{\text{after}}^C = \exp\left[ \frac{\ln \epsilon^2}{d_H(\delta_{\ell_1}, \delta_{\ell_{-1}})} \right]$$

When $\widehat{T} \le T_{\text{after}}^C$, then $\mathrm{TV}(p^{1,\widehat{T}}, p^1) \lesssim \epsilon$. When $\widehat{T} \ge T_{\text{before}}^C$, $\mathrm{TV}(p^{1,\widehat{T}}, p) \lesssim \epsilon$. For sufficiently large $d_H(\delta_{\ell_1}, \delta_{\ell_{-1}})$, the window size $T_{\text{before}}^C - T_{\text{after}}^C = O\left( \frac{1}{d_H(\delta_{\ell_1}, \delta_{\ell_{-1}})} \right)$. If $d_H(\delta_{\ell_1}, \delta_{\ell_{-1}})$ increases with $T$, then the width of the critical window compared to the width of the index set becomes negligible.

*Proof.* To prove $\mathrm{TV}(p^{1,\widehat{T}}, p^1) \lesssim \epsilon$ when $\widehat{T} \leq T^C_{\text{after}}$, observe that when $\widehat{T} \leq T^C_{\text{after}}$, the probability that all the differing elements between $\ell_1, \ell_{-1}$ are masked is exactly $\widehat{T}^{d_H(\delta_{\ell_1}, \delta_{\ell_{-1}})} \leq \epsilon^2$. That means that there exists a set $A$ with $p^{+1}_{\widehat{T}}(A) \geq 1 - \epsilon^2$ and $p^{-1}_{\widehat{T}}(A) = 0$, so by the definition of total variation, $\mathrm{TV}(p^{+1}_{\widehat{T}}, p^{-1}_{\widehat{T}}) \geq 1 - \epsilon^2$. Obviously, $\mathrm{TV}(p^{+1}_{\widehat{T}}, p^{+1}_{\widehat{T}}) = 0$ as well, so by Theorem 3.1, we obtain $\mathrm{TV}(p^{1,\widehat{T}}, p^1) \lesssim \epsilon$. To prove that $\mathrm{TV}(p^{1,\widehat{T}}, p^{\{\pm 1\}}) \leq \epsilon$ when $\widehat{T} \geq T^C_{\text{before}}$, we need only show that $\mathrm{TV}(p^{+1}_{\widehat{T}}, p_{\widehat{T}}) \leq \epsilon$. By Lemma 15 of (Li & Chen, 2024), it suffices to show that $\mathrm{TV}(p^{+1}_{\widehat{T}}, p^{-1}_{\widehat{T}}) \leq \epsilon$ by a simple triangle inequality argument. Consider the set $A \subset \{\pm 1, [\text{MASKED}]\}^T$ such that $\mathrm{TV}(p^{+1}_{\widehat{T}}, p^{-1}_{\widehat{T}}) = p^{+1}_{\widehat{T}}(A) - p^{-1}_{\widehat{T}}(A)$. Consider the set $B = \mathrm{supp}(p^{+1}_{\widehat{T}}) \cap \mathrm{supp}(p^{-1}_{\widehat{T}})$. For any $x \in B$, we know $p^{+1}_{\widehat{T}}(x) = p^{-1}_{\widehat{T}}(x)$ because the same number of tokens need to be masked from $\ell_{\pm 1}$. This means we have $p^{+1}_{\widehat{T}}(B) = p^{-1}_{\widehat{T}}(B) \geq \widehat{T}^{d_H(\ell_{+1}, \ell_{-1})} \geq 1 - \epsilon$. Because $p^{+1}_{\widehat{T}}(A) - p^{-1}_{\widehat{T}}(A) = p^{+1}_{\widehat{T}}(A - B) - p^{-1}_{\widehat{T}}(A - B)$, we have $p^{+1}_{\widehat{T}}(A) - p^{-1}_{\widehat{T}}(A) \leq p^{+1}_{\widehat{T}}(\{\pm 1, [\text{MASKED}]\}^T - B) \leq \epsilon$. $\square$

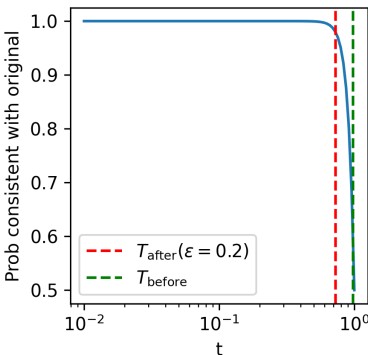

*Figure 6.* Plot of critical window for discrete diffusion on delta distributions with $T^C_{\text{before}}, T^C_{\text{after}}$ plotted.

## D.2. Autoregression

*Example* D.6. [Math problem-solving as a random walk] We model solving a math problem as taking a random walk on $\mathbb{Z}$ with stepsize 1 of length $T$. If the random walk hits $+A$, then it has 'solved' the problem; if the random walk hits $-A$, then it has obtained an incorrect solution. Assume that we have two modes: a strong problem solving mode (denoted $+1$), which takes a $+1$ step with probability $0.5 + \Delta$, and a weak problem solving mode (denoted $-1$), which takes a $+1$ step with probability $0.5 - \Delta$. Assuming that $\frac{\ln(2/\epsilon^2)}{2\Delta^2} < A$ and $\epsilon^2 < 10^{-3}(0.5 - \Delta)(0.5 + \Delta)$, there is a critical window for the strong problem solving window of $T^C_{\text{before}} = T - \frac{\epsilon^2}{\Delta^2} + 2$ and $T^C_{\text{after}} = T - \frac{\ln(2/\epsilon^2)}{2\Delta^2}$. Note the critical window has width $\Theta(1/\Delta^2)$ independent of $T$.

*Proof.* Because only the direction of steps matter, we can model the critical window for this random walk as observing a sequence of $\pm 1$ with an autoregressive language model. Let $p \in \{\pm 1\}^T$, and consider a forward process with index set $\mathbf{I} = \{0, 1, 2, \ldots, T\}$, $Y_0 = X$, and $Y_t \in \{\pm 1, [\text{MASKED}]\}^T$. For $t \in \mathbf{I}$, we let the last $t$ tokens of $Y_t$ be deterministically set to $[\text{MASKED}]$. We generate data as a mixture of biased coins with separation $2\Delta < 0.01$. For a mixture of two biased coins, with probabilities of $0.5 \pm \Delta$ ($\theta_{\pm 1}$ respectively) of yielding 1, we can compute the critical window and show that it tightly clusters around $\Theta(1/\Delta^2)$. Let $\Theta = \{\theta_{\pm 1}\}, p^{\theta_{\pm 1}} \sim (\text{Bern}(\theta_{\pm 1}))^{\otimes T}, w_{\pm 1} = \frac{1}{2}$. We also assume $\epsilon^2 < 10^{-3}(0.5 - \Delta)(0.5 + \Delta)$. Then, on component 1 we have the critical window $T^C_{\text{before}} = T - \frac{\epsilon^2}{\Delta^2} + 2$ and $T^C_{\text{after}} = T - \frac{\ln(2/\epsilon^2)}{2\Delta^2}$. When $\widehat{T} \leq T^C_{\text{after}}$, then $\mathrm{TV}(p^{1,\widehat{T}}, p^1) \lesssim \epsilon$. When $\widehat{T} \geq T^C_{\text{before}}$, $\mathrm{TV}(p^{1,\widehat{T}}, p) \lesssim \epsilon$.

Note that the number of $+1$ is sufficient for disambiguating $\theta_{\pm 1}$. To prove the bounds $T^C_{\text{before}}$, we show that with only $\frac{\epsilon^2}{\Delta^2} - 2$ samples the total variation between $0.5 - \Delta$ and $0.5 + \Delta$ is negligible. Using (Roos, 2001), we find

$$\mathrm{TV}(p^{+1}_{T^C_{\text{after}}}, p^{-1}_{T^C_{\text{after}}}) \leq \frac{2\Delta\sqrt{\frac{T - T^C_{\text{after}} + 2}{2(0.5 - \Delta)(0.5 + \Delta)}}}{\left(1 - 2\Delta\sqrt{\frac{T - T^C_{\text{after}} + 2}{2(0.5 - \Delta)(0.5 + \Delta)}}\right)^2} \lesssim 3\epsilon.$$

For $T_{\text{after}}^C$, we compute how many samples it takes for $p^{\pm 1}$ to have only $\epsilon$ overlap in total variation using Hoeffding's inequality. If we have $n$ samples, the mean $\overline{X}$ of the $n$ samples of $\pm 1$ for $p = 0.5 + \Delta$ satisfies the concentration inequality $P(|\overline{X} - 2\Delta| > \Delta) \leq 2\exp(-2nt^2)$ (furthermore we can ignore the stopping condition by our requirement that $\frac{\ln(2/\epsilon^2)}{2\Delta^2} < A$). We find $P(|\overline{X} - 2\Delta| > \Delta) \leq \epsilon^2$ for $T - T_{\text{after}}^C$ samples, proving that the total variation is at least $1 - \epsilon^2$. $\qquad\square$

Now, we consider a model for autoregressive data similar to the one presented in (Arora et al., 2019). Each word is a vector $w \in \mathbb{R}^d$ and the context length is $T \in \mathbb{Z}^{>0}$. The original samples are $x \in \mathbb{R}^{T \times d}$. Let $\Theta = \{u, v\}$, where $u, v \in \mathcal{S}^{d-1}$. We define the distribution $p^\theta$ for $\theta \in \Theta$ as follows. We generate the path of a discourse vector $\left(C_t^\theta\right)_{t \in [0,\infty]} \in \mathbb{R}^d$ with the reverse SDE Orstein-Uhlenbeck process such that $C_\infty^\theta \sim \mathcal{N}(0, \text{Id})$ and $C_0^\theta \sim \mathcal{N}(0, \text{Id} + \alpha\theta\theta^\top)$ for some $\alpha > 0$. We let $q_t^\theta$ be the law of $C_t^\theta$ for $t \geq 0$. We let $\mathbf{I} = \{0, 1, \ldots, T\}$, and for $t \in \mathbf{I}$, we draw samples $w_t^\theta \in \mathbb{R}^d$ where we impose a normal Gaussian prior and have $w_t^\theta | C_t^\theta \propto \exp(\langle C_t^\theta, \cdot \rangle)$. Then we return the corpus $\{w_t^\theta\}_{t \in \mathbf{I}}$ as an output.

**Theorem D.7** (Autoregressive with a mixture of two Gaussians as the concept distribution ). *We assume that* $\exp(-T)\sqrt{\alpha - \log(1 + \alpha)} \leq \epsilon$. *Let* $\phi(x) = \frac{2+x}{(2+x(1+\langle u,v \rangle)/2)(2+x(1-\langle u,v \rangle)/2)}$. *Then, on component $u$ we have*

$$T_{\text{after}}^C = \frac{1}{2}\ln\left[\frac{\alpha}{\phi^{-1}(\epsilon^2)}\right], T_{\text{before}}^C = \frac{1}{2}\ln\left[\frac{\alpha\sqrt{1 + \alpha^2}\sqrt{1 - \langle u,v \rangle^2}}{\epsilon}\right].$$

*When $\widehat{T} \leq T_{\text{after}}^C$, then $\text{TV}(p^{\{u\},\widehat{T}}, p^{\{u\}}) \lesssim \epsilon$. When $\widehat{T} \geq T_{\text{before}}^C$, $\text{TV}(p^{\{u\},\widehat{T}}, p) \lesssim \epsilon$.*

This proof will require Theorem D.8 and Lemmas D.2 and D.9, which is stated below:

**Theorem D.8** (Section 5.2 of (Chen et al., 2023)). *Let $(Y_t)_{t \in [0,T]}$ and $(Y_t')_{t \in [0,T]}$ denote the solutions to*

$$\begin{aligned} dY_t &= b_t(Y_t)\, dt + \sqrt{2}dB_t, & Y_0 &\sim q \\ dY_t' &= b_t'(Y_t')\, dt + \sqrt{2}dB_t, & Y_0' &\sim q. \end{aligned}$$

*Let $q$ and $q'$ denote the laws of $Y_T$ and $Y_T'$ respectively. If $b_t, b_t'$ satisfy that $\int_0^T \mathbb{E}_Q \|b_t(Y_t) - b_t'(Y_t)\|^2\, dt < \infty$, then* $\text{KL}(q\|q') \leq \int_0^T \mathbb{E}_Q \|b_t(Y_t) - b_t'(Y_t)\|^2\, dt.$

**Lemma D.9.** *Let $u, v \in \mathcal{S}^{d-1}$. Then*

$$\begin{aligned} \|uu^\top - vv^\top\|_{\text{op}} &\leq \sqrt{1 - \langle u,v \rangle^2} \\ \lambda(uu^\top + vv^\top) &= \{1 \pm \langle u,v \rangle\}. \end{aligned}$$

*Proof.* There exists $r \in \mathcal{S}^{d-1}$ such that $v = \langle u,v \rangle u + \sqrt{1 - \langle u,v \rangle^2}r$ and $u \perp r$. We find that

$$uu^\top - vv^\top = (1 - \langle u,v \rangle^2)uu^\top - (1 - \langle u,v \rangle^2)rr^\top - \langle u,v \rangle\sqrt{1 - \langle u,v \rangle^2}[ur^\top + ru^\top].$$

We can explicitly compute the eigenvalues of $uu^\top - vv^\top$ using the discriminant and find that they are equal to $\pm\sqrt{1 - \langle u,v \rangle^2}$. By a similar derivation, we can write

$$uu^\top + vv^\top = (1 + \langle u,v \rangle^2)uu^\top + (1 - \langle u,v \rangle^2)rr^\top + \langle u,v \rangle\sqrt{1 - \langle u,v \rangle^2}[ur^\top + ru^\top].$$

which gives us eigenvalues for $uu^\top + vv^\top$ of $1 \pm \langle u,v \rangle$. $\qquad\square$

*Proof.* To compute the $T_{\text{after}}^C$ bounds, we compare the difference in Hellinger distance of the distribution of words words at generated at index $\widehat{T}$, $w_{\widehat{T}}^u$. By the data processing inequality $1 - \frac{1}{2}H^2(p_{\widehat{T}}^u, p_{\widehat{T}}^v) \leq 1 - \frac{1}{2}H^2(w_{\widehat{T}}^u, w_{\widehat{T}}^v)$, so it suffices to show $1 - \frac{1}{2}H^2(w_{\widehat{T}}^u, w_{\widehat{T}}^v) \lesssim \epsilon$. Because the Gaussian is its own conjugate prior and $w_{\widehat{T}}^u | C_t^\theta \propto \exp(-\frac{1}{2}\|w_{\widehat{T}}^u - C_t^\theta\|^2)$, we can compute $w_{\widehat{T}}^u | C_t^\theta \sim \mathcal{N}(C_t^\theta, \text{Id})$ and $w_{\widehat{T}}^u \sim \mathcal{N}(0, 2\text{Id} + \alpha e^{-2t}uu^\top)$. Applying Lemmas D.2 and D.9, we can explicitly compute

$$1 - \frac{1}{2}H^2(w_{\widehat{T}}^u, w_{\widehat{T}}^v) \lesssim \sqrt{\frac{2 + \alpha e^{-2\widehat{T}}}{(2 + \alpha e^{-2\widehat{T}}(1 + \langle u,v \rangle)/2)(2 + \alpha e^{-2\widehat{T}}(1 - \langle u,v \rangle)/2)}} \lesssim \sqrt{\phi(\alpha e^{-2\widehat{T}})} \leq \epsilon.$$

To compute $T^C_{\text{before}}$, we first use the data processing inequality to reduce the difference in the emitted tokens to the difference in the paths of the context vectors, and then apply the approximation error bounds from Theorem D.8 to bound the differences in path measures. When $\widehat{T} \geq T^C_{\text{before}}$, we can use the triangle inequality to write $\text{TV}(p^{\{u\},\widehat{T}}, p) = \text{TV}(p^u_{\widehat{T}}, p^{\{u,v\}}_{\widehat{T}}) \leq \text{TV}(p^u_{\widehat{T}}, p^v_{\widehat{T}})$. Note that $p^\theta_{\widehat{T}}$ is the distribution of the first $T - \widehat{T}$ tokens generated by the model under $\theta$. Note that $p^u_{\widehat{T}}$ is a function of $(C^u_t)_{t \in \mathbf{I} \cap [\widehat{T}, T]}$ and $p^v_{\widehat{T}}$ is a function of $(C^v_t)_{t \in \mathbf{I} \cap [\widehat{T}, T]}$. By the data processing inequality, we can bound the difference in terms of the distributions over the tokens in terms of the law of the process of the discourse vectors,

$$\text{TV}(p^u_{\widehat{T}}, p^v_{\widehat{T}}) \leq \text{TV}((C^u_t)_{t \in \mathbf{I} \cap [\widehat{T}]}, (C^v_t)_{t \in \mathbf{I} \cap [\widehat{T}, T]}) \leq \text{TV}((C^u_t)_{t \in [\widehat{T}, T]}, (C^v_t)_{t \in [\widehat{T}, T]}).$$

Note that for $\theta \in \Theta$, $(C^\theta_t)_{t \in [\widehat{T}, T]}$ is generated by the following reverse time SDE,

$$dC^\theta_t = \{C^\theta_t + 2\nabla \ln q^\theta_t(C^\theta_t)\} \, dt + \sqrt{2} \, dB_t, \qquad t \in [\widehat{T}, T], C^\theta_T \sim q^\theta_T.$$

Now we define $(C^{u \to v}_t)_{t \in [\widehat{T}, T]}$ to be the reverse SDE defined by initializing at $q^u_T$ but with the score of $q^v_t$,

$$dC^{u \to v}_t = \{C^{u \to v}_t + 2\nabla \ln q^v_t(C^{u \to v}_t)\} \, dt + \sqrt{2} \, dB_t, \qquad t \in [\widehat{T}, T], C^{u \to v}_T \sim q^u_T.$$

By the triangle inequality, we have

$$\text{TV}((C^u_t)_{t \in [\widehat{T}, T]}, (C^v_t)_{t \in [\widehat{T}, T]}) \leq \underbrace{\text{TV}((C^u_t)_{t \in [\widehat{T}, T]}, (C^{u \to v}_t)_{t \in [\widehat{T}, T]})}_{\text{(I)}} + \underbrace{\text{TV}((C^{u \to v}_t)_{t \in [\widehat{T}, T]}, (C^v_t)_{t \in [\widehat{T}, T]})}_{\text{(II)}}.$$

To bound (I), observe that the SDEs have different scores but the same initializations. We apply Theorem D.8 to $\text{TV}((C^u_t)_{t \in [\widehat{T}, T]}, (C^{u \to v}_t)_{t \in [\widehat{T}, T]})$ and obtain

$$\text{TV}((C^u_t)_{t \in [\widehat{T}, T]}, (C^{u \to v}_t)_{t \in [\widehat{T}, T]}) \leq \sqrt{\text{KL}((C^u_t)_{t \in [\widehat{T}, T]} \| (C^{u \to v}_t)_{t \in [\widehat{T}, T]})}$$

$$\leq \sqrt{\int_{\widehat{T}}^T \mathbb{E}_{X \sim C^u_t} \|\nabla \ln p^u_t(X) - \nabla \ln p^v_t(X)\|^2 dt}$$

We simplify the inner expectation by using the

$$\|\nabla \ln p^u_t(X) - \nabla \ln p^v_t(X)\| = \| \left[ (\text{Id} + \alpha e^{-2t} uu^\top)^{-1} - (\text{Id} + \alpha e^{-2t} vv^\top)^{-1} \right] x \|$$

$$= \| \left[ \left( \text{Id} - \frac{\alpha e^{-2t}}{1 + \alpha e^{-2t}} uu^\top \right) - \left( \text{Id} - \frac{\alpha e^{-2t}}{1 + \alpha e^{-2t}} vv^\top \right) \right] x \|$$

$$\lesssim \alpha e^{-2t} \|uu^\top - vv^\top\|_{\text{op}} \left\| \prod_{\text{span}(u,v)} x \right\|,$$

$$= \alpha e^{-2t} \sqrt{1 - \langle u, v \rangle^2} \left\| \prod_{\text{span}(u,v)} x \right\|. \quad \text{(Lemma D.9)}$$

We can upper bound $\mathbb{E}_{X \sim C^u_t} \| \prod_{\text{span}(u,v)} X \|^2$ by considering right-triangular $L$ such that $L^\top L = \text{Id} + \alpha uu^\top$. and $\prod_{\text{span}(u,v)} X = \prod_{\text{span}(u,v)} LY$, where $Y \sim \mathcal{N}(0, \text{Id})$. The operator norm of $\prod_{\text{span}(u,v)} L$ is

$$\left\| \prod_{\text{span}(u,v)} \circ L \right\|_{\text{op}} \leq \|L\|_{\text{op}} \leq \sqrt{1 + \alpha^2 e^{-2t}} \leq \sqrt{1 + \alpha^2}.$$

$\prod_{\text{span}(u,v)} \circ L$ is also rank 2 and $\left( \prod_{\text{span}(u,v)} \circ L \right) Y = \left( \prod_{\text{span}(u,v)} \circ L \right) \circ \prod_{L^{-1} \text{span}(u,v)} Y$, where $\prod_{L^{-1} \text{span}(u,v)} Y \sim \mathcal{N}(0, \text{Id}_2)$. Thus we have

$$\mathbb{E}_{Y \sim \mathcal{N}(0, \text{Id})} \left[ \left\| \left( \prod_{\text{span}(u,v)} \circ L \right) Y \right\|^2 \right] = \mathbb{E}_{Y' \sim \mathcal{N}(0, \text{Id}_2)} \left[ \left\| \left( \prod_{\text{span}(u,v)} \circ L \right) Y' \right\|^2 \right] \lesssim 1 + \alpha^2$$

Combining this information together, we are able to compute,

$$\text{TV}((C_t^u)_{t\in[\widehat{T},T]}, (C_t^{u\to v})_{t\in[\widehat{T},T]}) \leq \sqrt{\int_{\widehat{T}}^{T} \mathbb{E}_{X\sim C_t^u} \|\nabla \ln p_t^u(X) - \nabla \ln p_t^v(X)\|^2 dt}$$

$$\lesssim \alpha\sqrt{1+\alpha^2}\sqrt{1-\langle u,v\rangle^2}e^{-2\widehat{T}}$$

$$\lesssim \epsilon.$$

To bound (II), we observe that both are run with the same score so we need only bound the difference at initialization. By the data processing inequality, we again have $\text{TV}((C_t^{u\to v})_{t\in[\widehat{T},T]}, (C_t^v)_{t\in[\widehat{T},T]}) \leq \text{TV}(p_T^u, p_T^v)$. We can again apply the triangle inequality to get $\text{TV}(p_T^u, p_T^v) \leq \text{TV}(p_T^u, \gamma^d) + \text{TV}(\gamma^d, p_T^v) \leq .$ For any $\theta \in \Theta$, we have by the forward convergence of the OU process $\text{TV}(p_T^u, \gamma^d) \leq \exp(-T)\sqrt{\text{KL}(p_u||\gamma^d)}$. We can explicitly compute $\text{KL}(p_u||\gamma^d)$ as

$$\text{KL}(p_u||\gamma^d) = \frac{1}{2}\left[d + \alpha - d - \log(\text{Id} + \alpha uu^\top)\right] = \frac{1}{2}\left[\alpha - \log(1+\alpha)\right].$$

Thus, we obtain the following bound on $\text{TV}((C_t^{u\to v})_{t\in[\widehat{T},T]}, (C_t^v)_{t\in[\widehat{T},T]})$ of

$$\text{TV}((C_t^{u\to v})_{t\in[\widehat{T},T]}, (C_t^v)_{t\in[\widehat{T},T]}) \lesssim \exp(-T)\sqrt{\alpha - \log(1+\alpha)} \lesssim \epsilon.$$

$$\square$$

## D.3. Interweaving transitions from other distributions

In this section, we extend our critical windows framework to the setting where at certain steps of sampling procedure, instead of using the reverse Markov transition kernel from the original stochastic localization sampler, we use an alternative distribution which is not necessarily related to the original sampler. This includes many important applications of generative models, in which one seeks to combine the priors learned from data with some other algorithm. For example, one may want to combine the language model with a problem generation oracle in in-context learning (Dong et al., 2024).

As (Xie et al., 2022) points out, the transition from the answer to one problem to the problem statement of another example in-context learning is determined by an alternative transition kernel (which they call $p_{\text{prompt}}$). Although the probability of transition from one answer to the problem statement of another example is extremely low under the natural data distribution, one still hopes that with sufficiently many samples, the model selects the correct $\theta^* \in \Theta$ if these lower probability transitions are overcome by the distributional difference for $\theta \in \Theta$ with $\theta \neq \theta^*$. Similarly, under our critical windows framework, we can hope to capture the idea that we specialize to a particular $\theta^*$ given a sufficiently long context. In Section D.3.1, we first present a general framework for characterizing critical windows in this setting. Then, in section D.3.2, we consider the case of in-context learning by autoregressive language models and prove convergence.

### D.3.1. GENERAL INTERWEAVING FRAMEWORK

We present this framework for the case where the index set $\mathbf{I} = \{0, 1, \ldots, m\}$ is discrete. Like before, assume we have a series of reverse Markov transition kernels $P_{k\to k-1}^{\leftarrow,p}(\cdot|\cdot)$, for $k \in \mathbf{I}$, but we also assume we have an alternative distribution $P_{k\to k-1}^{\leftarrow,\text{alt}}(\cdot|\cdot)$ that we use to sample for transitions $k \in A \subsetneq \mathbf{I}$. For our sampling procedure, we sample $Y_m$, and for $k = m-1, m-2, \ldots, 0$, we take $Y_k \sim P_{k+1\to k}^{\leftarrow,p}(\cdot|Y_{k+1})$ for $k \in \mathbf{I} - A$ and $Y_k \sim P_{k+1\to k}^{\leftarrow,\text{alt}}(\cdot|Y_{k+1})$ for $k \in A$. We denote the final distribution $p^{\text{alt}}$.

Now, we also need to adjust our definitions of $p^S$ to this particular sampling procedure. We define $p^{S,\text{alt}}$ for $S \subset \Theta$ to the distribution over outputs when we instead use the kernels $P_{k+1\to k}^{\leftarrow}(\cdot|Y_{k+1}, S)$ instead of $P_{k+1\to k}^{\leftarrow}(\cdot|Y_{k+1})$. To relate $p^{\theta,\text{alt}}$ to $p^{\theta,\text{alt}}$ for $\theta \in \Theta$, we need to assume transitions from alt do not affect the posterior distribution over $p^\Theta(\theta|Y_t)$.

**Assumption D.10.** For all $y \in \mathbb{R}$ and $x \in \text{supp}(P_{k+1\to k}^{\leftarrow,\text{alt}}(\cdot|y))$, we have for all $\theta, \theta' \in \Theta$, the equality $P_{k+1\to k}^{\leftarrow}(x|y, \theta) = P_{k+1\to k}^{\leftarrow}(x|y, \theta')$.

Adopting our definitions from Section 3, we let

$$T_{\text{lower,alt}}(\epsilon) \in \{t \in \mathbf{I} : \text{TV}(p_t^{S_{\text{init}},\text{alt}}, p_t^{S_{\text{targ}},\text{alt}}) \leq \epsilon\}$$
$$T_{\text{upper,alt}}(\epsilon) \in \{t \in \mathbf{I} : \text{TV}(p_t^{S_{\text{targ}},\text{alt}}, p_t^{\Theta - S_{\text{targ}},\text{alt}}) \geq 1 - \epsilon^2\}.$$

The main challenge of the below corollary is simply show that the final distribution $p^{\Theta,\text{alt}}$ can be written as a mixture of $p^{\theta,\text{alt}}$ with the same mixing weights as before.

**Corollary D.11.** *Under Assumption D.10, for $\epsilon > 0$, if $\widehat{T} \geq T_{\text{lower,alt}}(\epsilon)$ and $\widehat{T} \leq T_{\text{upper,alt}}(\epsilon)$, then*

$$\text{TV}(p^{S_{\text{init}},\widehat{T}}, p^{S_{\text{targ}},\text{alt}}) \leq \left(1 + \sqrt{2} \max\left(1, \frac{\sum_{\theta \in \Theta - S_{\text{targ}}} w_\theta}{\sum_{\theta \in S_{\text{targ}}} w_\theta}\right)/2\right)\epsilon.$$

*Proof.* We need only show that $p^{\Theta,\text{alt}} \triangleq \sum_{\theta \in \Theta} w_\theta p^{\theta,\text{alt}}$. It suffices to shows that the probability of generating a path $Y_m, Y_{m-1}, \ldots, Y_0$ are the same under both density functions. We need only consider transitions for $k \in \mathbf{I} - A$, because for $k \in A$, the transitions are both given by the alternative distribution. For the transitions not given by alt, note that we are using the original model, so

$$p^{\Theta,\text{alt}}(Y_{k-1}|Y_k) = \frac{\sum_{\theta \in \Theta} w_\theta p^\theta(Y_k) P_{k \to k-1}^{\leftarrow}(Y_{k-1}|Y_k, \theta)}{\sum_{\theta \in \Theta} w_\theta p^\theta(Y_k)}.$$

Furthermore, for the mixture model, this probability is

$$p^{\text{mix}} = \frac{\sum_{\theta \in \Theta} w_\theta p^{\theta,\text{alt}}(Y_m, Y_{m-1}, \ldots, Y_{k+1}, Y_k) P_{k \to k-1}^{\leftarrow}(Y_{k-1}|Y_k, \theta)}{\sum_{\theta \in \Theta} w_\theta p^{\theta,\text{alt}}(Y_m, Y_{m-1}, \ldots, Y_{k+1}, Y_k)}.$$

The distinction between Equation D.3.1 and Equation D.3.1 is that in the former we are using the likelihood of $p^\theta$ instead of $p^{\theta,\text{alt}}$. Thus it suffices to show that $p^\theta \propto p^{\theta,\text{alt}}$. We explicitly write out the probability,

$$p^\theta(Y_m, Y_{m-1}, \ldots, Y_{k+1}, Y_k) = \prod_{i=k+1}^{m} p^\theta(Y_{i-1}|Y_i) \propto \prod_{i=k+1, i \notin A}^{m} p^\theta(Y_{i-1}|Y_i),$$

where the proportionality follows from the fact that we can ignore the probability of the transitions produced by alt under Assumption D.10. By definition, this is proportional to $p^{\theta,\text{alt}}(Y_m, Y_{m-1}, \ldots, Y_{k+1}, Y_k)$ up to a normalization constant independent of $\theta$. $\square$

### D.3.2. IN-CONTEXT LEARNING

Now, we will specialize our framework to the case of in-context learning. As in (Xie et al., 2022), we assume that the language model is given inputs of the form $[x_1, y_1, o, x_2, y_2, o, \ldots, x_t, y_t, o, x_{t+1}]$, where $x_1$ is the input, $y_1$ is the output, and $o$ is a delimiter token that separate different in-context samples from each other. We assume that the transitions $y_i \to o \to x_{i+1}$ are sampled by some alternative probability distribution $P_{\text{other}}(\cdot|\cdot)$. We require that $P_{\text{other}}$ selects the $x_i$ i.i.d.

**Assumption D.12.** The distribution of $P_{\text{other}}(x_{t+1}|x_1, y_1, o, x_2, y_2, o, \ldots, x_t, y_t, o) = P_{\text{other}}(x_1)$.

Then we assume that the transitions $x_i \to y_i$ are generated by some $\theta^* \in \Theta$, which does not depend on any of the previous tokens before the delimiter.

**Assumption D.13.** (Well-specification) There exists some $\theta^* \in \Theta$ such that $y_i$ is generated from $y_i \sim P^{\leftarrow}(\cdot|[x_1, y_1, o_1, \ldots, o_{i-1}, x_i], \theta^*)$.

**Assumption D.14.** For all $\theta \in \Theta$, we have $P^{\leftarrow}(\cdot|[x_1, y_1, o_1, \ldots, o_{i-1}, x_i], \theta) = P^{\leftarrow}(\cdot|x_i, \theta)$.

We also assume statistical separation of $\theta^*$ from $\Theta - \{\theta^*\}$ in terms of Hellinger distance.

**Assumption D.15.** Let $p_{(x,y)}^S$ for $S \subset \Theta$ be the distribution of $(x_1, y_1, o)$ where $x_1 \sim P_{\text{other}}(\cdot|o)$ and $y_1 \sim p^S(\cdot|x_1)$. There exists $\delta > 0$ such that $H^2(p_{(x,y)}^{\Theta - \{\theta^*\}}, p_{(x,y)}^{\theta^*}) \geq \delta$.

*Example* D.16. Let $T \geq \ln\left(\frac{1-\delta/2}{\epsilon}\right)$. Under Assumptions D.10, D.12, D.13, D.14, and D.15, we have $\mathrm{TV}([x_1, \ldots, x_{T+1}, y_{T+1}], [x_1, \ldots, x_{T+1}, \tilde{y}_{T+1}]) \lesssim \epsilon/w_{\theta^*}$.

*Proof.* It suffices to upper bound $1 - \frac{1}{2}H^2(P_{3T}^{\theta,\mathrm{alt}}, P_{3T}^{\Theta-\{\theta\},\mathrm{alt}})$ by $O(\epsilon)$. First observe that the distribution $p^{S,\mathrm{alt}}$ for $S \subset \Theta$ factors along the delimiters by a factor independent of $S$ using Assumptions D.12 and D.14, so we have

$$P_{3t}^{S,\mathrm{alt}}(x_1, y_1, o_1, x_2, y_2, o_2, \ldots, x_t, y_t, o_t) \propto \prod_{i=1}^{t} P_3^{S,\mathrm{alt}}(x_i, y_i, o_i).$$

Using the tensorization property of Hellinger distance and our definition of $T, \delta$, we have

$$1 - \frac{1}{2}H^2(P_{3T}^{\theta,\mathrm{alt}}, P_{3T}^{\Theta-\{\theta\},\mathrm{alt}}) \lesssim \left[1 - \frac{1}{2}H^2(P_3^{\theta,\mathrm{alt}}, P_3^{\Theta-\{\theta\},\mathrm{alt}})\right]^T \leq \epsilon.$$

$\square$

## D.4. All-or-nothing

### D.4.1. ALL-OR-NOTHING PHENOMENON

Here we elucidate a formal connection between the critical windows phenomenon in in-context learning and the *all-or-nothing* phenomenon. To begin, we first define the notions of strong and weak detection:

**Definition D.17.** Let $(N_s)$ be an increasing sequence of integers. Given sequences of distributions $(p_s), (q_s)$ over $z \in \mathbb{R}^{N_s}$, a sequence of test statistics $(\mathcal{A}_s : \mathbb{R}^{N_s} \to \mathbb{R})$ with threshold $(\tau_s)$ achieves:

- *strong detection* if $\limsup_{s\to\infty}\{\Pr_{z\sim p_s}[\mathcal{A}_s(z) < \tau_s] + \Pr_{z\sim q_s}[\mathcal{A}_s(z) \geq \tau_s]\} = 0$.

- *weak detection* if $\limsup_{s\to\infty}\{\Pr_{z\sim p_s}[\mathcal{A}_s(z) < \tau_s] + \Pr_{z\sim q_s}[\mathcal{A}_s(z) \geq \tau_s]\} < 1$.

By the operational characterization of TV distance, strong detection is (information-theoretically) possible if and only if $\liminf_{s\to\infty} \mathrm{TV}(p_s, q_s) = 1$, and weak detection is (information-theoretically) possible if and only if $\liminf_{s\to\infty} \mathrm{TV}(p_s, q_s) > 0$.

Now we consider the following Bayesian inference problem, given by a joint distribution $\pi$ over $(\theta, z) \in \mathbb{R}^n \times \mathbb{R}^m$. Nature samples unknown signal $\theta \in \mathbb{R}^n$ from $\pi_\theta$. Given sample size $N$, we receive observations $\{z_i\}_{i=1}^N$ drawn i.i.d. from $\pi_{z|\theta}$; the goal is to infer $\theta$ from these observations. Let $\pi^{(N)}$ denote the distribution over $\{z_i\}_{i=1}^N$, the mixture of product measures parametrized by $\theta$.

**Definition D.18.** Let $(\pi_s)$ be a sequence of inference tasks over $\mathbb{R}^{n_s} \times \mathbb{R}^{m_s}$ and $(\pi^{\mathrm{null}_s})$ be a sequence of distributions over $\mathbb{R}^{m_s}$. $(\pi_s)$ exhibits an *all-or-nothing phase transition at threshold* $(N_s)$ *with respect to null models* $(\pi^{\mathrm{null}_s})$ if:

- For any $\beta < 1$: weak detection between $(\pi^{(\beta N_s)})$ and $((\pi^{\mathrm{null}_s})^{\otimes \beta N_s})$ is information-theoretically impossible
- For any $\beta > 1$: strong detection between the planted model $(\pi^{(\beta N_s)})$ and the null model $((\pi^{\mathrm{null}_s})^{\otimes \beta N_s})$ is information-theoretically possible

All-or-nothing phase transitions have been established for a number of natural inference tasks like sparse linear regression (Reeves et al., 2019; Gamarnik & Zadik, 2019), sparse PCA (Niles-Weed & Zadik, 2020), generalized linear models (Barbier et al., 2020), group testing (Truong et al., 2021; Coja-Oghlan et al., 2022), linear and phase retrieval models (Scarlett & Cevher, 2016; Truong & Scarlett, 2020), planted subgraphs (Mossel et al., 2023), and planted Gaussian perceptron (Niles-Weed & Zadik, 2023). Here is an example for sparse linear regression:

**Theorem D.19** ((Reeves et al., 2019)). *Let $\pi_s$ be the distribution over $\mathbb{R}^{n_s} \times \mathbb{R}^{m_s}$ for $n_s = s$ and $m_s = s + 1$ where the marginal over $\theta$ is given by the uniform distribution over $k_s$-sparse vectors in $\{0,1\}^s$, and the conditional distribution $\pi_{z|\theta}$ is given by sampling $x \sim \mathcal{N}(0, \mathrm{Id}_s)$, taking $y = \langle \theta, x \rangle + \xi$ for $\xi \sim \mathcal{N}(0, \sigma_s^2)$, and outputting observation $z = (x, y)$. The null model $\pi_s^{\mathrm{null}}$ is given by sampling $x \sim \mathcal{N}(0, \mathrm{Id}_s)$ and outputting $y = \mathcal{N}(0, k_s + \sigma_s^2)$.*

*If $\sigma_s^2 \ll k_s \leq s^{0.499}$, then $(\pi_s)$ exhibits an all-or-nothing phase transition at threshold $(N_s^*)$ with respect to null models $(\pi_s^{\mathrm{null}})$ for $N_s^* \triangleq \frac{2k_s \log(s/k_s)}{\log(1+k_s/\sigma_s^2)}$.*

Having defined the all-or-nothing phenomenon, we rigorously instantiate it as a critical window for in-context learning. We first define a mixture model $p_{(N)}^{\Theta}$ for sequence lengths $N$ onto which we will identify a critical window.

**Definition D.20.** To any inference task $\pi$, null model $\pi^{\mathsf{null}}$, and sequence length $N$, we can associate the following in-context learning task. Let $\Theta = \Theta_{\mathsf{signal}} \sqcup \{\mathsf{NULL}\}$ where $\Theta_{\mathsf{signal}} \triangleq \mathrm{supp}(\pi_\theta)$. Given $\theta \in \mathrm{supp}(\pi_\theta)$, let $p_{(N)}^{\theta}$ denote the distribution over sequences $(z_1, \ldots, z_N, ?, \theta)$ where $z_1, \ldots, z_N$ are i.i.d. samples from $p_{z|\theta}$. Let $p_{(N)}^{\mathsf{null}}$ denote the distribution over observations $(z_1, \ldots, z_N, ?, \mathsf{NULL})$ where $z_1, \ldots, z_N$ are i.i.d. samples from $\pi^{\mathsf{null}}$. We then take $p_{(N)}^{\Theta} \triangleq \mathbb{E}_{\theta \sim \frac{1}{2}\pi_\theta + \frac{1}{2}\delta_{\mathsf{NULL}}} p_{(N)}^{\theta}$.

Under this model of data, we have the following theorem expressing the all-or-nothing phase transition in terms of $T_{\mathrm{before}}^{C}, T_{\mathrm{after}}^{C}$.

**Theorem D.21.** *Suppose $(\pi_s)$ is a sequence of inference tasks that exhibits an all-or-nothing phase transition at threshold $(N_s^*)$ with respect to null models $(\pi_s)$. Given $N_s \geq N_s^*$, let $(p_{(N_s)}^{s;\theta})_{\theta \in \Theta_s}$ denote the sequence of in-context learning tasks. For any constant $0 < \epsilon < 1$, there exist constants $\delta, \underline{s}$ such that for all $s \geq \underline{s}$, next-token prediction for $(p_{(N_s)}^{s;\theta})_{\theta \in \Theta_s}$ exhibits a critical window over $[N_s + 2 - (1+\delta)N_s^*, N_s + 2 - -(1-\delta)N_s^*]$ in which we transition from sampling a distribution $O(\epsilon)$-close in TV to $S_{\mathrm{before}} = \Theta_{s;\mathsf{signal}}$, to sampling from a distribution $O(\epsilon)$-close in TV to $S_{\mathrm{after}} = \Theta_s$.*

*In other words, we have $T_{\mathrm{before}}^{C} \triangleq N + 2 - (1-\delta)N_s^*$ and $T_{\mathrm{after}}^{C} \triangleq N + 2 - (1+\delta)N_s^*$.*

The proof of Theorem D.21 is essentially immediate from Theorem 3.1 and the definition of the all-or-nothing phase transition:

*Proof.* Let us first apply Theorem 3.1 to $S_{\mathrm{init}} = S_{\mathrm{targ}} = \Theta_{s;\mathsf{signal}}$. By the definition of $D_\Theta$, the parameter $W$ therein is 1. Furthermore, we trivially have that $T_{\mathrm{end}}(\epsilon) = 0$. Finally, because strong detection is possible provided there are $N \geq \beta N_s^*$ in-context examples for $\beta > 1$, there exists $\delta_1$ depending only on $\epsilon$ for which $\mathrm{TV}(p_t^{S_{\mathrm{targ}}}, p_t^{\Theta_s - S_{\mathrm{targ}}}) \geq 1 - \epsilon^2$ for $t = N_s + 2 - (1+\delta_1)N_s^*$. By Theorem 3.1 we conclude that $\mathrm{TV}(p^{S_{\mathrm{init}}, N_s + 2 - (1+\delta_1)N_s^*}) \lesssim \epsilon$. Next, let us apply Theorem 3.1 to $S_{\mathrm{init}} = \Theta_{s;\mathsf{signal}}$ and $S_{\mathrm{targ}} = \Theta_s$. The parameter $W$ therein is now 0. Furthermore, we trivially have that $T_{\mathrm{end}}(\epsilon) = N_s + 2$. Finally, because weak detection is impossible provided there are $N \leq \beta N_s^*$ in-context examples for $\beta < 1$, there exists $\delta_2$ depending only on $\epsilon$ for which $\mathrm{TV}(p_t^{S_{\mathrm{init}}}, p_t^{S_{\mathrm{targ}}}) \leq \epsilon$ for $t = N_s + 2 - (1-\delta_2)N_s^*$. By Theorem 3.1 we conclude that $\mathrm{TV}(p^{S_{\mathrm{init}}, N_s + 2 - (1-\delta_2)N_s^*}) \leq \epsilon$. Taking $\delta = \max(\delta_1, \delta_2)$ concludes the proof. $\square$

# E. Details from Section 5

**Definition E.1.** An $\epsilon$-**mixture tree** is a tuple $(T, \{P^{\rightarrow}(\cdot|\cdot)\}, \mathbf{I}, \Theta, \{p^\theta\}_{\theta \in \Theta}, \mathrm{Subset}, \mathrm{NoiseAmount})$. The tree $T = (V, E)$ is associated with a function $\mathrm{Subset}: V \rightarrow 2^{\Theta} \backslash \{\emptyset\}$, which maps vertices to sub-mixtures. We require Subset satisfies the following two properties: (1) $\mathrm{Subset}(\mathrm{root}) = \Theta$; (2) If $u$ is a parent of $v$, $\mathrm{Subset}(v) \subset \mathrm{Subset}(u)$. We consider a $\mathrm{NoiseAmount}: V \rightarrow \mathbb{R}^{\geq 0}$, which characterizes the noise levels that result in the aggregations of mixture components described by vertices in the mixture tree. Thus we require that NoiseAmount satisfy three properties: (1) For distinct $\theta_i, \theta_j \in \Theta$ with leaf nodes $w, v$ such that $\theta_i \in \mathrm{Subset}(w), \theta_j \in \mathrm{Subset}(v)$, if $u$ is the lowest common ancestor of $w, v$, then we require $\mathrm{TV}(p_{\mathrm{NoiseAmount}(u)}^{\theta_i}, p_{\mathrm{NoiseAmount}(u)}^{\theta_j}) \leq \epsilon$; (2) For $u \in V$, we have statistical separation between $\mathrm{Subset}(u)$ and $\Theta - \mathrm{Subset}(u)$ in terms of TV, $\mathrm{TV}(p_{\mathrm{NoiseAmount}(u)}^{\mathrm{Subset}(u)}, p_{\mathrm{NoiseAmount}(u)}^{\Theta - \mathrm{Subset}(u)}) \geq 1 - \epsilon^2$; and (3) If $v \in V$ is a parent of $u$, we have $\mathrm{NoiseAmount}(u) < \mathrm{NoiseAmount}(v)$. Property 1 establishes bounds on $T_{\mathrm{end}}^{S}$, and properties 2 and 3 establishes bounds on $T_{\mathrm{st}}^{S}$.

**Corollary E.2.** *Consider an $\epsilon$-mixture tree. For $\theta_i \in \Theta$, consider the path $u_1, u_2, u_3, \ldots, u_{H'} \in V$ where $u_1$ is the leaf node with $\theta_i \in \mathrm{Subset}(u_1)$ and $u_{H'}$ is the root. There is a sequence of times $T_1 < T_2 < \cdots < T_{H'}$ with $\mathrm{TV}(p^{\{i\}, T_\ell}, p^{\mathrm{Subset}(u_\ell)}) \lesssim_w \epsilon$.*

*Proof.* For $\ell \in [H']$, we let $T_\ell = \mathrm{NoiseAmount}(u_\ell)$. We apply Theorem 3.1 with $S_{\mathrm{init}} = \{i\}$ and $S_{\mathrm{targ}} = \mathrm{Subset}(u_\ell)$. We know $\mathrm{TV}(p_{T_\ell}^{S_{\mathrm{targ}}}, p_{T_\ell}^{\Theta - S_{\mathrm{targ}}}) \geq 1 - \epsilon^2$ by Condition 2 in Definition E.1. By Lemma 15 of (Li & Chen, 2024), we know $\mathrm{TV}(p_{T_\ell}^{\{i\}}, p_{T_\ell}^{S_{\mathrm{init}}}) \leq \max_{j \in S_{\mathrm{init}}} \mathrm{TV}(p_{T_\ell}^{\{i\}}, p_{T_\ell}^{\{j\}})$. This is $\leq \epsilon$ for all $j \in S_{\mathrm{init}}$ by Condition 3 on NoiseAmount and the data processing inequality. $\square$

**Corollary E.3.** *Consider an $\epsilon$-mixture tree $(T, \{P^{\rightarrow}(\cdot|\cdot)\}, \mathbf{I}, \Theta, \{p^{\theta}\}_{\theta \in \Theta}, Subset, NoiseAmount)$. Suppose we have another distribution $\{q^{\theta}\}_{\theta \in \Theta}$ such that $\text{TV}(p^{\theta}, q^{\theta}) \leq \delta/2$ for all $\theta \in \Theta$. Then we have $\epsilon + \sqrt{\delta}$-mixture tree given by $(T, \{P^{\rightarrow}(\cdot|\cdot)\}, \mathbf{I}, \Theta, \{q^{\theta}\}_{\theta \in \Theta}, Subset, NoiseAmount)$.*

*Proof.* We need only check the first and second properties of NoiseAmount with parameter $\epsilon + \sqrt{\delta}$. To do this, it suffices to show $\text{TV}(q^{\theta_i}_{\text{NoiseAmount}(u)}, q^{\theta_j}_{\text{NoiseAmount}(v)}) \leq \epsilon + \delta$ and $\text{TV}(q^{\text{Subset}(u)}_{\text{NoiseAmount}(u)}, q^{\Theta - \text{Subset}(u)}_{\text{NoiseAmount}(u)}) \geq 1 - \epsilon^2 - \delta$. By the data processing inequality, we just need to show this at $t = 0$, and we prove the stronger statement that for $S_1 \subset \Theta$, $\text{TV}(p^{S_1}, q^{S_1}) \leq \delta/2$. This follows from Lemma 15 of (Li & Chen, 2024) and $\text{TV}(p^{\theta}, q^{\theta}) \leq \delta/2$ for all $\theta \in \Theta$. $\qquad\square$

*Example* E.4. Consider a set of alphabets $\{\mathcal{A}_i\}_{i=1}^d$ and define $\Theta = \{(a_i)_{i=1}^d : \forall i \in [d], a_i \in \mathcal{A}_i\}$ and $p^{\theta_i} = \delta_{\theta_i}$. Let $\mathbf{I} = [0, 1, 2, \ldots, d]$. and for any permutation $i_1, i_2, \ldots, i_d$ of $[d]$, define a forward process such that at $t \in \mathbf{I}$, we mask all $i_d, i_{d-1}, \ldots, i_{d-t}$. This constructs a hierarchy where the values for $i_1, i_2, \ldots, i_d$ are decided in that order.

*Proof.* We construct the following 0-mixture tree as follows. We let the leaf nodes be the set $\Theta$. We let two leaf nodes $u, v$ have the same parent if and only if they share the same values on the alphabet at $i_1, i_2, \ldots, i_{d-1}$; we also define the parent as the union of all of its children. We now treat the parents we constructed as the roots, and let them have the same parent if and only if they share the same values on the tuple $i_1, i_2, \ldots, i_{d-2}$. We continue to do this until we are left with one root node. We let Subset map each node to the corresponding set and NoiseAmount map each node to its distance from a leaf node.

By the construction of $T$, it is clear that Subset satisfies the desired properties. For distinct $\theta_i, \theta_j \in \Theta$, the lowest common ancestor of $\theta_i, \theta_j$ represents the largest $k$ such that indices $i_1, \ldots, i_k$ are the same for $\theta_i, \theta_j$. Because $p^{\theta_i}_{\text{NoiseAmount}(u)}$ is just the tuple of the values of $\theta_i, \theta_j$ at $i_1, \ldots, i_k$, we know $\text{TV}(p^{\theta_i}_{\text{NoiseAmount}(u)}, p^{\theta_j}_{\text{NoiseAmount}(u)}) = 0$. For any $u \in V$ representing the values at index $(i_\ell)_{\ell=1}^k$, all $\theta \notin \text{Subset}(u)$ does not share the same values at these indices by definition, so we also know

$$\text{TV}(p^{\text{Subset}(u)}_{\text{NoiseAmount}(u)}, p^{\Theta - \text{Subset}(u)}_{\text{NoiseAmount}(u)}) = 1.$$

$\qquad\square$

## F. Jailbreak Experiments

### F.1. Reproducing critical windows for jailbreaks from existing papers

Existing work has already identified the presence of critical windows in the domain of jailbreaks. Here we present critical windows for a simplified prefill jailbreak based on the prefill attack (Haize Labs, 2024b) and repeating token jailbreak (Nasr et al., 2023) for LLAMA-3.1-8B-Instruct. In the first figure, we plot the probability of the model giving a harmful response, computed using the `StrongReject` Gemma 7b auditor from (Souly et al., 2024), as a function of the fraction of the phrase `Sure, here is how to` appended to the front of the model's generation. We can see that there is a large jump in the attack success rate after only including a few tokens in the prefix. The second figure is a reproduction of Figure 12 from (Nasr et al., 2023). It shows that the probability of repeating the next token increases substantially as the first few tokens are included a few times in the context.

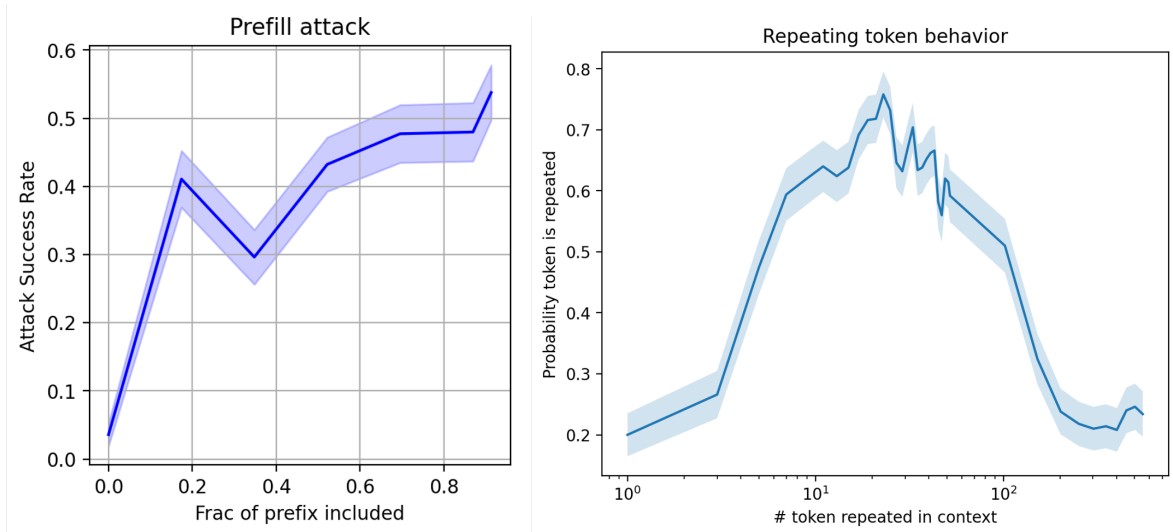

*Figure 7.* Examples of critical windows in jailbreaks for LLAMA-3.1-8B-Instruct. **Left:** Critical window for a prefill jailbreak (Haize Labs, 2024b). **Right:** Critical window for a repeating token jailbreak (Nasr et al., 2023).

### F.2. Experimental details from jailbreak

Now we apply our theory to develop a new jailbreak detection method, based on a likelihood ratio between an aligned and unaligned model. Intuitively, our theory states that when the unaligned component assigns a high probability to the text compared to the entire model, the model is likely to be jailbroken. We use a LLAMA-3.1-8B-Instruct model jailbroken with LoRA to not refuse harmful prompts (grimjim, 2024) as a proxy for the unaligned model. We evaluate these different methods on a dataset of jailbreaks and benign prompts from (Bailey et al., 2024).

**Dataset.** We use the same dataset as (Bailey et al., 2024) but provide details here for completeness. The benign dataset consists of inputs from UltraChat (Ding et al., 2023), a large dialogue dataset, and Xstest (Röttger et al., 2024), which contains benign queries that are often incorrectly refused by language models. The benign queries are filtered to ensure that LLAMA-3.1-8B-Instruct does not refuse any of them. The dataset of harmful prompts is based off of the Circuit Breakers dataset (Zou et al., 2024). The datasets include the following jailbreaking methods from the extant literature: PAIR (Chao et al., 2023), AutoDAN (Liu et al., 2024), Many-Shot Jailbreaking (MSJ) (Anil et al., 2024), Multi-Turn Attacks (Li et al., 2024; Haize Labs, 2024a), Prefill, GCG (Zou et al., 2023), and other Misc. attakcs from (Wei et al., 2023). For each jailbreaking method, it is applied to a prompt from the Circuit Breaker dataset and evaluated to see if the generation from LLAMA-3.1-8B-Instruct is helpful and harmful, as determined by the StrongReject jailbreaking classifier (Souly et al., 2024)).

**Evaluation Metric.** As is standard in the jailbreak detection literature (Bailey et al., 2024), we report the recall at the false positive rate at 0.01.

Table 2 displays the recall and several other baselines. Crucially, the log likelihood ratio methods does obtain recall > 1 for 5 different categories of jailbreaks. While our methods do perform worse than existing methods, it is important to note that they still work and that their poor performance could be explained by the fact that we have to use a proxy for the unaligned mode of the model.

*Table 2.* Recall (FPR=0.01) for our likelihood ratio threshold, a perplexity threshold (Alon & Kamfonas, 2023), and a MLP-based detector trained on activations (Bailey et al., 2024) for predicting different jailbreaks. prompt/gen denote the logprobs of the prompt and generation, respectively.

| | AutoDAN | GCG | Multi-Turn | Misc | MSJ | Pair | Prefill |
|---|---|---|---|---|---|---|---|
| $\log p_{\text{prompt}}^{\text{unaligned}} - \log p_{\text{prompt}}^{\text{aligned}}$ | 0.000 | 0.000 | 0.028 | 0.000 | 0.063 | 0.000 | 0.077 |
| $\log p_{\text{gen}}^{\text{unaligned}} - \log p_{\text{gen}}^{\text{aligned}}$ | 0.082 | 0.030 | 0.000 | 0.100 | 0.000 | 0.061 | 0.051 |
| $\log p_{\text{prompt}}^{\text{aligned}}$ | 0.000 | 0.576 | 0.056 | 0.063 | 0.013 | 0.000 | 0.077 |
| $\log p_{\text{gen}}^{\text{aligned}}$ | 0.205 | 0.150 | 0.570 | 0.200 | 0.006 | 0.015 | 0.416 |
| MLP | 1.00 | 0.956 | 0.873 | 0.663 | 1.00 | 0.833 | 1.00 |

# G. Structured output experiments

We have LLAMA-3.1-8B-Instruct (default sampling parameters of temperature of $0.6$ and top-p sampling of $0.9$) respond to following prompt, which asks it to answer a series of fill-in-the-blank questions in a structured format. We prefill the model's generations with `\n\n 1.` to ensure that the outputs comport to that format. Figure 8 plots the probability of obtaining the same answers as the original generation after truncating different amounts from the generation in the forward-reverse experiments, computed with $10,000$ generations from the prompt. Our theory predicts that jumps in the probability will occur at $T_{\text{before}}^C, T_{\text{after}}^C$ which represent when the model has committed to a particular answer to a question in the generation. To compute $T_{\text{before}}^C, T_{\text{after}}^C$, we look at when the generations diverge based on the first occurrence of the identifying information. For example, the $T_{\text{before}}^C$ of the first critical window is `1.   The `, because the first answer has not appeared in the generation, and the $T_{\text{after}}^C$ of the first critical window is `1.   The P` or `1.   The N`, because that uniquely identifies the answer. These theoretical predictions are vindicated empirically, as the jumps in probability, representing the model localization to a particular set of answers, occur exactly at $T_{\text{before}}^C, T_{\text{after}}^C$ in Figure 8.

---

**Structured Output Prompt**

Complete the following by choosing only one option for each blank. The options are provided in parentheses, and your response must match the exact case and meaning of the chosen option. Respond with only the completed sentence, no explanations or additional text.
1. The (Pirate/Ninja) jumped across the ship.
2. She adopted a (Dog/Cat) from the shelter.
3. The (River/Bridge) sparkled under the sun.
4. A (Dragon/Knight) guarded the castle gates.
5. He ordered (Pizza/Sushi) for dinner.

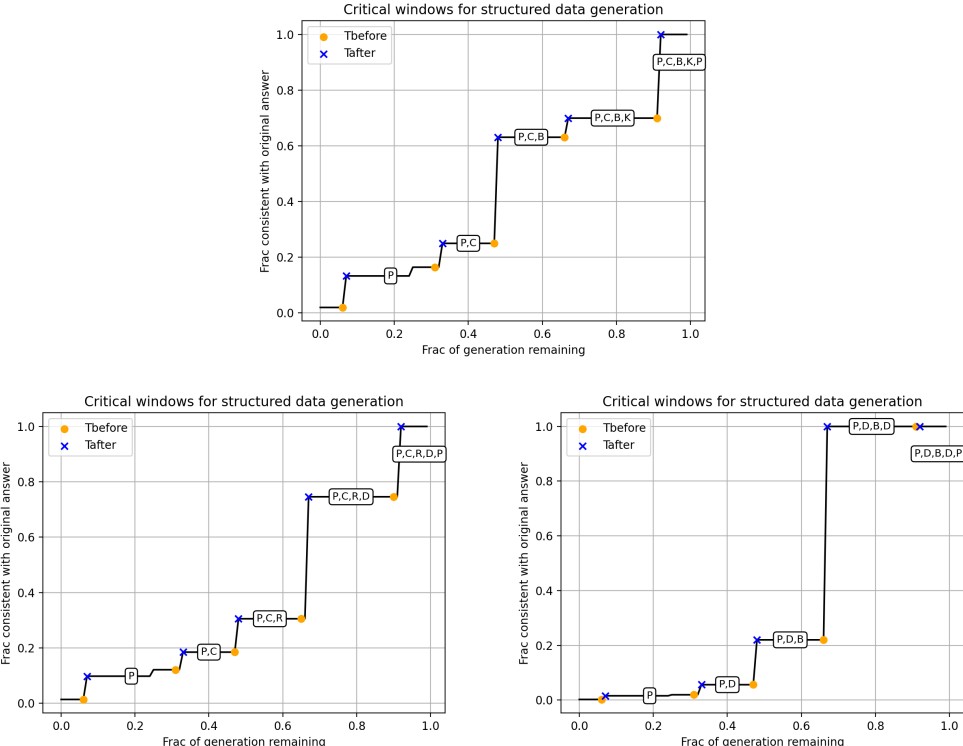

*Figure 8.* Structured output plots for LLAMA-3.1-8B-Instruct. P denotes that we are sampling from responses whose answer to the first question was Pirate; P,C denotes that we are sampling from responses whose answers to the first two questions were Pirate and Cat, respectively. We can see that the critical windows directly correspond to our theoretical values for $T_{\text{before}}^C, T_{\text{after}}^C$.

## H. Chain of thought experiments

### H.1. Experimental details

We select 7 different math and reasoning benchmarks on which performance is known to improve with chain of thought reasoning (Lanham et al., 2023): ARC Challenge and Easy (Clark et al., 2018), AQua (Ling et al., 2017), LogiQA (Liu et al., 2020), MMLU (Hendrycks et al., 2021a), and TruthfulQA (Lin et al., 2022) multiple-choice benchmarks and the MATH benchmark from (Hendrycks et al., 2021b).

We describe the prompts we used in our evaluation of different datasets. The system prompt for all datasets is `Produce a correct solution to the following /TASK/ question.`, where `/TASK/` is the type of question of the dataset, i.e. science, math, or logic. For each question, we create a user prompt by appending `Think of the /TASK/ question thoroughly step by step.  Please only respond with the answer after reasoning thoroughly.` in front of the question. Once the model completes its generation (max generation length set to 2048 and default sampling parameters), we append the user prompt `Given all of the above, what's the single, most likely answer?  Your answer should have the format "The answer is ANSWER", where ANSWER is your answer.` for the multiple choice benchmarks and `Given all of the above, what's the single, most likely answer?  Simplify it completely. Your answer should have the format "The answer is $ANSWER$", where ANSWER is your answer in LaTeX.` Note that when we ask the model for the final answer, we set the temperature to 0.

For 400 questions from each dataset, we prompt the language model with the query, cut off the generation at different percentages of completion (in tokens), and regenerate it 100 times. We compute the probability that the final answer is the same as the original answer, using a direct text comparison for the multiple choice benchmarks and the prm800k grader for MATH (Lightman et al., 2023). Across three different models ( LLAMA-3.1-8B-Instruct, Phi-3-7B-Instruct, and Qwen-

*Table 3.* Differences between Accuracy (Acc) without versus with critical windows and frequency of critical windows (CW) when the original generation is wrong versus right for LLAMA-3.1-8B-Instruct, Phi-3-7B-Instruct, and Qwen-2.5-7B-Instruct.

| Dataset | LLAMA-3.1-8B-Instruct | | Phi-3-7B-Instruct | | Qwen-2.5-7B-Instruct | |
|---|---|---|---|---|---|---|
| | $\Delta$**Acc** | $\Delta$**CW** | $\Delta$**Acc** | $\Delta$**CW** | $\Delta$**Acc** | $\Delta$**CW** |
| AQUA-RAT | 0.42 | 0.20 | 0.36 | 0.16 | 0.03 | 0.01 |
| ARC Challenge | 0.53 | 0.22 | 0.50 | 0.24 | 0.38 | 0.11 |
| ARC Easy | 0.73 | 0.26 | 0.28 | 0.13 | 0.40 | 0.07 |
| LogiQA | 0.15 | 0.11 | 0.21 | 0.19 | 0.23 | 0.11 |
| MATH | 0.41 | 0.33 | 0.36 | 0.33 | 0.46 | 0.29 |
| MMLU | 0.45 | 0.24 | 0.38 | 0.21 | 0.26 | 0.11 |
| TruthfulQA | 0.44 | 0.20 | 0.42 | 0.23 | 0.55 | 0.23 |

2.5-7B-Instruct), we find that conditioned on the occurrence of critical windows, the model generation is significantly less accurate compared to generations without critical windows.

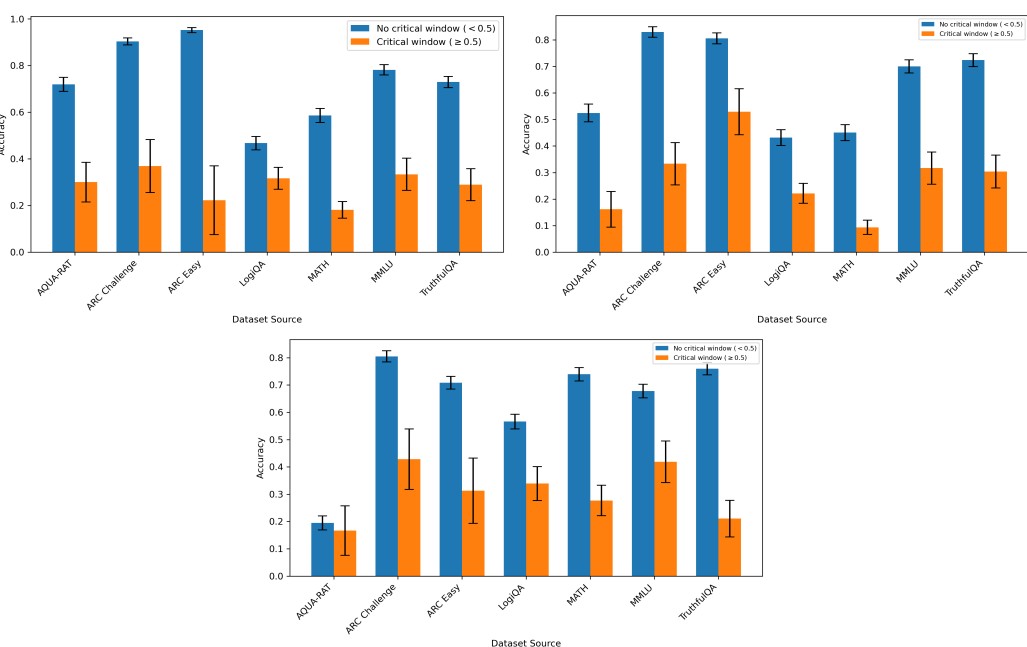

*Figure 9.* Left-to-right: LLAMA-3.1-8B-Instruct, Phi-3-7B-Instruct, and Qwen-2.5-7B-Instruct barplots of original generation accuracy for generations with ($\geq 0.5$ jump in probability) and without critical windows ($< 0.5$).

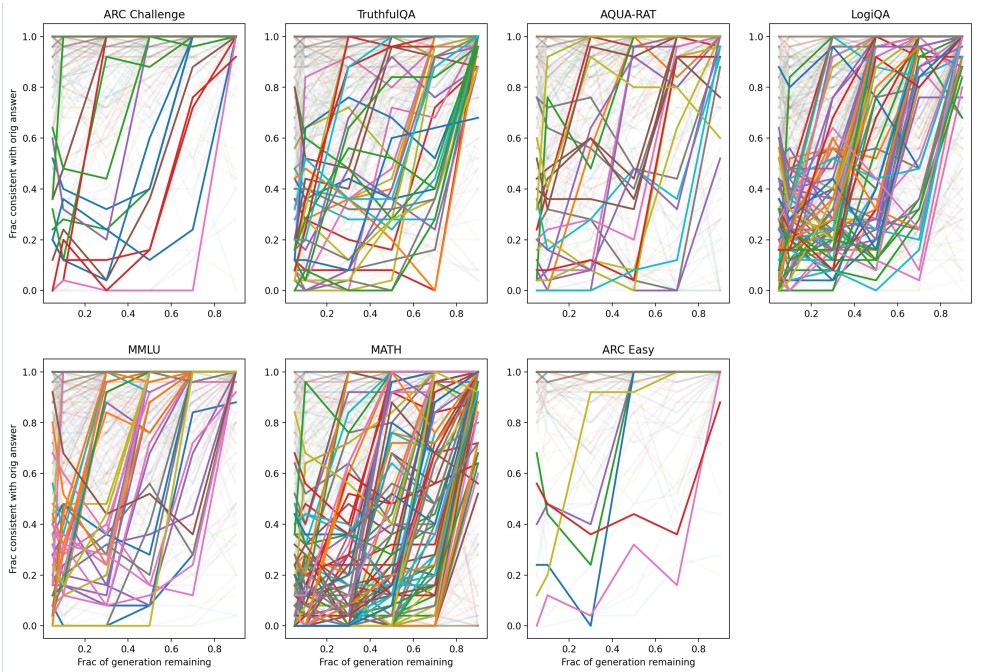

*Figure 10.* Probability that the answer is the same as a function of the percentage of the generation remaining for different math and reasoning benchmarks for LLAMA-3.1-8B-Instruct. Highlighted are generations with a 50% increase and no 30% decrease in the probability over one step.

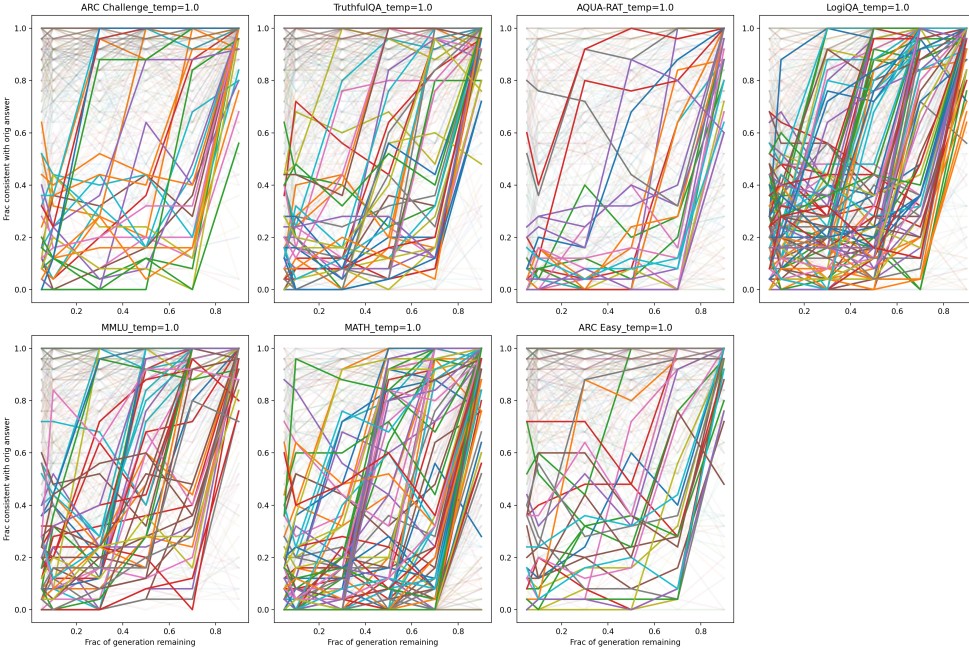

*Figure 11.* Probability that the answer is the same as a function of the percentage of the generation remaining for different math and reasoning benchmarks for Phi-3-7B-Instruct. Highlighted are generations with a 50% increase and no 30% decrease in the probability over one step.

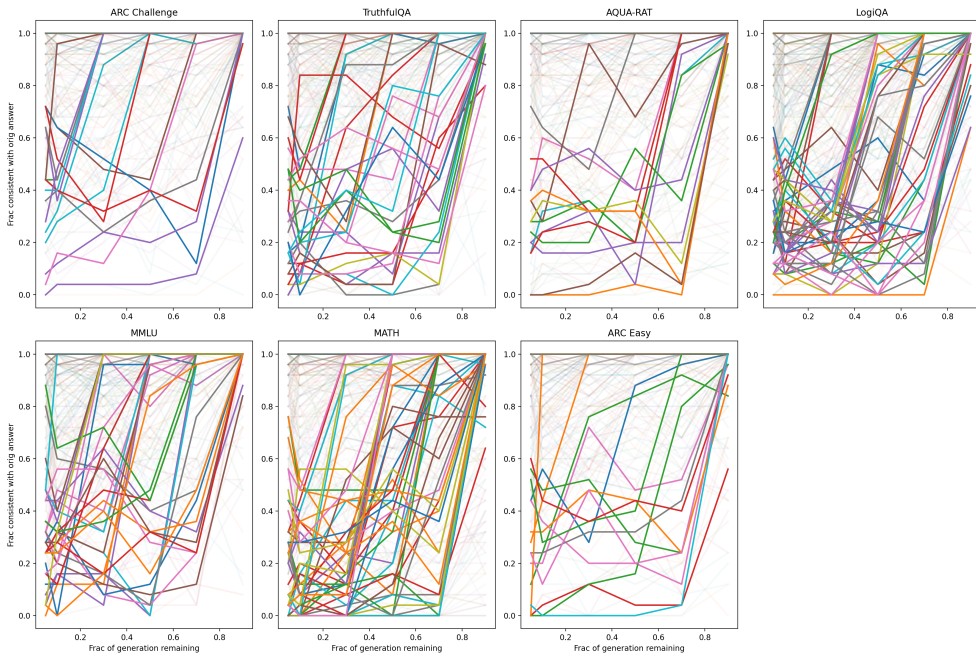

*Figure 12.* Probability that the answer is the same as a function of the percentage of the generation remaining for different math and reasoning benchmarks for Qwen-2.5-7B-Instruct. Highlighted are generations with a 50% increase and no 30% decrease in the probability over one step.

## H.2. Examples of critical windows

We provide a few more examples of critical windows (bolded).

---

**MATH example (LLAMA-3.1-8B-Instruct)**

**Q**: Let $a$ and $b$ be the solutions of the equation $2x^2 - 10x + 5 = 0$. What is the value of $(2a - 3)(4b - 6)$?

**A**: To find the value of $(2a - 3)(4b - 6)$, we need to first find the values of $a$ and $b$ by solving the equation $2x^2 - 10x + 5 = 0$. We can use the quadratic formula to solve for $a$ and $b$: $x = \frac{-b \pm \sqrt{b^2 - 4ac}}{2a}$. In this case, $a = 2$, $b = -10$, and $c = 5$. Plugging these values into the quadratic formula, we get: $x = \frac{-(-10) \pm \sqrt{(-10)^2 - 4(2)(5)}}{2(2)}$ $x = \frac{10 \pm \sqrt{100 - 40}}{4}$ $x = \frac{10 \pm \sqrt{60}}{4}$ $x = \frac{10 \pm 2\sqrt{15}}{4}$ So, the solutions are $a = \frac{10 + 2\sqrt{15}}{4}$ and $b = \frac{10 - 2\sqrt{15}}{4}$. Now that we have the values of $a$ and $b$, we can substitute them into the expression $(2a - 3)(4b - 6)$: $(2a - 3)(4b - 6) = (2(\frac{10 + 2\sqrt{15}}{4}) - 3)(4(\frac{10 - 2\sqrt{15}}{4}) - 6)$ Simplifying the expression, we get: $(2a - 3)(4b - 6) = (\frac{10 + 2\sqrt{15}}{2} - 3)(\frac{10 - 2\sqrt{15}}{1} - 6)$ $(2a - 3)(4b - 6) = (\frac{10 + 2\sqrt{15} - 6}{2})(\frac{10 - 2\sqrt{15} - 6}{1})$ $(2a - 3)(4b - 6) = (\frac{4 + 2\sqrt{15}}{2})(\frac{4 - 2\sqrt{15}}{1})$ **Now, we can simplify the expression further by multiplying the numerators and denominators:** $(2a - 3)(4b - 6) = \frac{(4 + 2\sqrt{15})(4 - 2\sqrt{15})}{2}$ **Using the difference of squares formula** $(a + b)(a - b) = a^2 - b^2$**, we can simplify the numerator:** $(2a - 3)(4b - 6) = \frac{4^2 - (2\sqrt{15})^2}{2}$ $(2a - 3)(4b - 6) = \frac{16 - 60}{2}$ Simplifying the expression further, we get: $(2a - 3)(4b - 6) = \frac{-44}{2}$ $(2a - 3)(4b - 6) = -22$ Therefore, the value of $(2a - 3)(4b - 6)$ is $-22$.

**Correct answer**: $-22$

---

---

LogiQA example (Phi-3-7B-Instruct)

**Q**: The person in charge of the relevant department said that pilots will be carried out in some areas this year to prepare for the system and policy preparations for the comprehensive clean-up of "small property houses", and require all localities to confirm the registration and certification of rural collective land. All small-property houses are not subject to registration and are not protected by law. Therefore, this newly built house in Hexi Village is not protected by law. Which of the following is true will most weaken the above argument. (A) This newly built house in Hexi Village has been tacitly approved by relevant departments. (B) This newly built house in Hexi Village is a small property house. (C) This newly built house in Hexi Village is built on rural collective land. (D) Some of the newly built houses in Hexi Village are not small property houses.

**A**: The statement implies that small property houses are not protected by law and can be demolished if necessary in the preparations for the system and policy preparations for the comprehensive clean-up. -We have to find the option that weakens this argument the most. Option A suggests that the department has tacitly approved the newly built house. This weakens the argument, **but it can still be argued that the house is subject to demolition if the comprehensive clean-up policy is implemented. Option B states that this newly built house in Hexi Village is a small property house. This directly contradicts the argument that all small-property houses are not protected by law** and not subject to registration and certification. Therefore, this house would not be demolished and would be protected by law. Option C says that the house is built on rural collective land. This doesn't disprove the argument – it might still be a small-property house without legal protection. Option D makes only some of the newly built houses in Hexi Village not small property houses. This doesn't address the argument that all small-property houses are not protected by law and not subject to registration, but only carries an exceptional circumstance. Thus, the option that most weakens the argument is (B) as it directly contradicts the statement by implying that the specific house is protected by law. The answer is B.

**Correct answer**: D

---

MATH example (Qwen-2.5-7B-Instruct)

**Q**: Carson flips over the cards of a standard 52-card deck one at a time. What is the probability that he flips over the ace of spades before any face card (jack, queen or king)?

**A**: 1. First, we need to identify the relevant cards for our problem. The ace of spades (A♠) and the face cards (J, Q, K) are the ones we're focusing on. There are 4 face cards in a standard deck of 52 cards (12 face cards total, but we only consider 4 for this problem). 2. There are 52 cards in total, and 5 of these (A♠, J♠, Q♠, K♠, J♡, Q♡, K♡, J♢, Q♢, K♢, J♣, Q♣, K♣) are either the ace of spades or a face card. Therefore, there are 47 other cards that are neither the ace of spades nor face cards. 3. The problem can be simplified by considering the probability of the ace of spades coming before any face card in a sequence of drawing these 9 cards (the ace of spades and the **8 face cards). 4. We can think of the problem as a sequence of drawing 9 cards, where we are interested in the position of the ace of spades relative to the face cards. The ace of spades can come in any of the 9 positions, and we want to find the probability that it comes before any of the face cards. 5. For the ace of spades to come before any** face card, it must be in one of the first 5 positions (since there are 4 face cards, and the ace of spades must come before any of them). There are 9 positions in total, and 5 of these are favorable. 6. Therefore, the probability that the ace of spades comes before any face card is the ratio of the number of favorable outcomes to the total number of outcomes, which is $\frac{5}{9}$. So, the probability that Carson flips over the ace of spades before any face card is $\boxed{\dfrac{5}{9}}$.

**Correct answer**: $\frac{1}{13}$

## H.3. Varied temperature sampling

In our above experiments with critical windows, we have been generating model responses at their default sampling temperatures: LLAMA-3.1-8B-Instruct: 0.6, Phi-3-7B-Instruct: 1.0, and Qwen-2.5-7B-Instruct: 0.7. A natural question to ask is whether the sampling temperature affects the presence or strength of critical windows across these models. At one extreme, sampling at infinite temperature exhibits no critical windows since each token is generated uniformly at random. At the other extreme, zero temperature sampling is deterministic, so there is only one mode of behavior and critical windows again cannot exist. At intermediate temperatures, decreasing sampling temperature reduces token variance, so the sharpness of transition from general mixture to target submixture can be heightened. At the same time, reduced sampling variance also reduces the likelihood of transitioning between modes as well, leaving the overall effect on critical windows unclear.

First, we report the frequency of critical windows across the different temperatures for the MATH dataset: $0.1, 0.4, 0.7$, and $1.0$. Then, we report the average size of a critical window on $400$ samples from MATH, running the forward-reverse experiment on each sample $25$ times and taking the max consecutive jump as the critical window size for that problem. From Figures 13 and 14, each model has its own relationship between temperature and critical window frequency and size, with no clear definite relationship across all models.

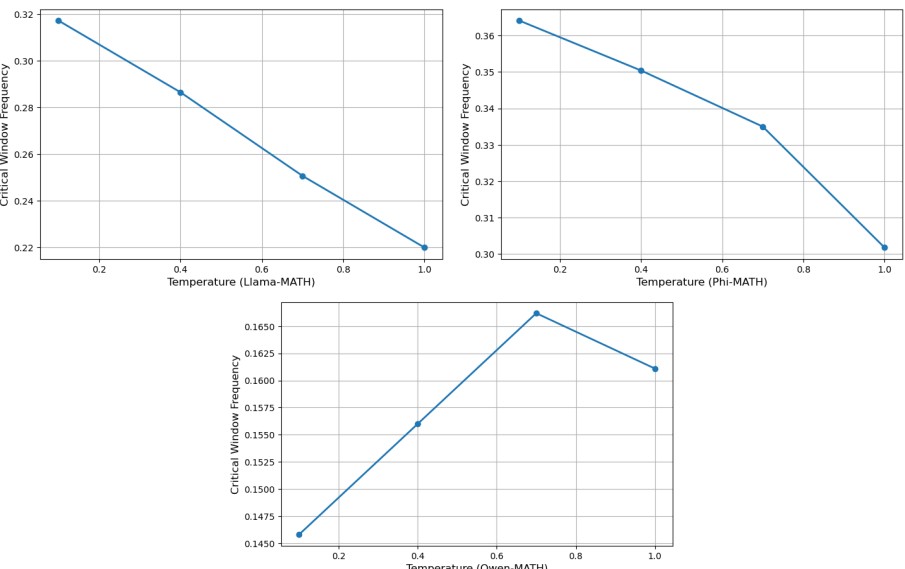

*Figure 13.* Temperature versus critical window frequency for LLAMA-3.1-8B-Instruct, Phi-3-7B-Instruct, and Qwen-2.5-7B-Instruct.

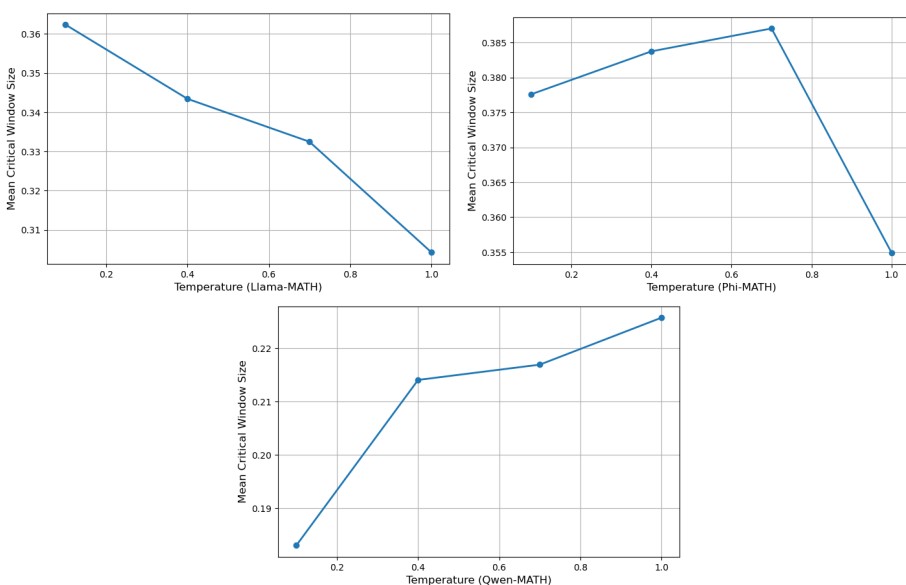

*Figure 14.* Temperature versus mean critical window size for LLAMA-3.1-8B-Instruct, Phi-3-7B-Instruct, and Qwen-2.5-7B-Instruct.

