# OpenReview forum: "Blink of an eye: a simple theory for feature localization in generative models"
_ICML.cc/2025/Conference — ICML 2025 oral_

### Official Review · Reviewer_MfZY · 2025-03-09

**Overall Recommendation:** 4

**Summary:**

This paper introduces a general framework for critical windows in stochastic localization. After a lengthy but valuable description of some key notions such as stochastic localization sampling and the "forward-reverse experiment", the authors prove their key result, which shows that there exist (possibly empty) "critical windows" during stochastic localization sampling in which the forward process has destroyed the information needed to distinguish a submixture $S_{init}$ from a larger submixture $S_{targ}$, but not the information needed to distinguish $S_{targ}$ from the remainder of the data distribution. They provide a simple toy example in the case of diffusion models and a number of examples drawn from autoregressive learning. They then sketch a general theory of how stochastic localization interacts with hierarchical semantic structure and briefly describe the results of some experiments on LLMs.

## Update after rebuttal.

I appreciate the authors' engagement with my review and their promise to add figures and examples to the camera-ready as intuition-building tools. I maintain my positive assessment of this paper.

**Claims And Evidence:**

The main claim in this paper is to have constructed a general theory to explain "critical windows" in generative models, which the authors colloquially define as a small subset of steps in which important features of a model sample emerge. I believe the paper largely achieves this objective. The stochastic localization framework is general enough to include diffusion models and autoregressive models, which are the two main paradigms in generative modeling nowadays. Their main result is general and does not rely on strong assumptions or particularly heavy sledgehammer results. However, as the authors acknowledge in Remark 3.2, their theory does not rule out the possibility of critical windows being empty sets, which I believe to be the main gap in their story. Nonetheless, I think the results are sufficiently interesting to stand on their own, and look forward to future work exploring why critical windows are often non-empty for common classes of generative models.

**Essential References Not Discussed:**

While I am not well-versed in the literature on critical windows and stochastic localization, it seems to me that this paper adequately situates itself in its literature.

**Experimental Designs Or Analyses:**

The experiments included in the main body seem to be sound and adequately illustrate that critical windows can occur in LLMs. However, I would have liked the authors to include a more thorough discussion of their experiments in the main body -- as I will note below, it generally seems like the authors have packed too much content into the 8-page limit at the price of e.g. a very abbreviated experiments section.

**Methods And Evaluation Criteria:**

The theoretical methods are appropriate for demonstrating the authors' key claims. In particular, the "forward-reverse experiment" is an appropriate tool for formalizing the process of destroying and then recovering information in stochastic localization sampling.

**Other Comments Or Suggestions:**

In general, it seems like the authors attempted to pack too much content into the 8-page limit, and have consequently neglected to include illustrative figures and a related work section in the main body of the text. They have also heavily compressed their experiments section. If this paper is accepted, I'd ask the authors to consider including figures, an expanded experiments section, and at least an abbreviated related work section in the main body, perhaps at the price of moving some of the examples in Section 4 to the appendix.

**Other Strengths And Weaknesses:**

While this paper is fairly well-written, it is dense with notation and short on figures. A few figures to illustrate the key notions would greatly improve the readers' intuition for the results. For example, the authors could include a figure illustrating the forward-reverse experiment for a simple case like a mixture of Gaussians and depicting the model distribution $p_t^S$ for various subsets $S$ during the critical window predicted by Theorem 3.1. The definition of $\epsilon$-mixture trees in Section 5 is also *very* abstract, and while I appreciate the benefits of this approach from the standpoint of generality, a few figures or additional examples would help readers parse the definitions better.

To me, the most interesting takeaway from this paper is that despite knowing nothing about the semantics of the data a priori, a stochastic localization sampler generates information in a way that respects the data's semantic hierarchy. e.g. a dog is an animal, so the support of the distribution over images of dogs is contained in the support of the distribution over images of animals -- and stochastic localization features critical windows in which the sampler has "decided" to generate an animal image but not yet decided that it will generate a dog image. It seems surprising to me that one can prove a general result of this form. However, I believe that Remark 3.2 reveals the primary gap in this theory as it stands -- it is not clear a priori that non-empty critical windows should exist. Exploring *why* this is the case would be an interesting future direction.

**Questions For Authors:**

No further questions.

**Relation To Broader Scientific Literature:**

This paper generalizes results from Li and Chen (2024), which studies the existence of critical windows in diffusion models. It draws heavily on tools from the stochastic localization literature, which is anchored by a series of papers by Eldan and connected to diffusion models in a set of notes by Montanari (2023).

**Theoretical Claims:**

I have reviewed the proof outline for Theorem 3.1. I believe the strategy is correct, and I was unable to find any specific errors in the outline.

---

> ### Author Rebuttal · Authors · 2025-04-01
>
> We would like to thank the reviewer for their time and thoughtful comments. We were glad to hear that you thought that the theoretical results were interesting and that experiments were sound and illustrative of our main points.
>
> ## Writing changes
>
> * *“While this paper is fairly well-written, it is dense with notation and short on figures”*
>
> In the final revision, we will include a section titled “Intuition for critical windows” before the technical prelims that **describes our theory informally** and introduces the forward-reverse experiment, the definition of critical windows, the definition of $\epsilon$-mixture trees, and our main Theorem 3.1 through a **simple vignette**. All of these definitions and theorems will be accompanied by **figures that visually explain them and text which concretely places them within our vignette.** For example, the definition of the forward-reverse experiment will be accompanied by a figure which shows how a “sweet spot” of noise leads to the specialization to a target sub-mixture; the definition of $\epsilon$-mixture trees will be shown with a graph that shows the hierarchy of features in our vignette; we will expand and add more detail to Figure 2 of a critical window in this section.
>
> The section will very loosely follow this structure: we will describe critical windows as the transition from sampling from a larger subset of features to a smaller subset of features. This motivates trying to understand when the generative model is sampling from a subset of features, and thus the forward-reverse experiment and our main Theorem 3.1. We will provide intuition into the location of the bounds for Theorem 3.1 and then explain how sequences of critical windows motivate understanding a hierarchy of feature specialization and thus the definition of $\epsilon$-mixture trees.
>
> * *“For example, the authors could include a figure illustrating the forward-reverse experiment...”*
>
> Yes, in the aforementioned new section we will illustrate the forward-reverse experiment with a very concrete example and figure.
>
> * *"The definition of ϵ-mixture trees in Section 5 is also very abstract, and while I appreciate the benefits of this approach from the standpoint of generality, a few figures or additional examples would help readers parse the definitions better."*
>
>
> We also plan to include an example and figure of an “$\epsilon$-mixture tree” in the section “Intuition for critical windows” accompanying our text and the vignette.
>
> * *“include a more thorough discussion of their experiments in the main body… consider including figures, an expanded experiments section, and at least an abbreviated related work section in the main body”*
>
> In addition to the new section, we will move many details from the examples and hierarchy sections to the appendix, add a short related works section in the main body, and thoroughly expand the experiments section. The expanded experiments section will include our structured output experiments, which show that our theory is predictive of critical windows for LLMs when outputs are hierarchically structured, and we will add to our LLM reasoning experiments details that were originally relegated to the appendix, e.g., statistics and visualizations of critical windows across datasets and models. The abbreviated related works section will cover the theory of critical windows for diffusion, the forward-reverse experiment, and stochastic localization.
>
> ## Future directions
> * *“not clear a priori that non-empty critical windows should exist. Exploring why this is the case would be an interesting future direction.”*
>
> In the Yellowstone and jailbreaking example, some actions from the LLM, i.e. browsing Yellowstone or acceding to a harmful user request, are much likelier under one mode of behavior than another and completely determine to which mode the generation belongs, resulting in a critical window as explained by Example 4.3. We agree with the reviewer that further exploring why critical windows exist in different settings is an interesting direction of future research.
>
> Thank you again for your time in reviewing the paper and providing much helpful feedback. If we have addressed your concerns about the paper, we hope you consider raising our score.

---

### Official Review · Reviewer_aK3T · 2025-03-10

**Overall Recommendation:** 4

**Summary:**

This paper discusses the phenomenon of critical windows in generative models. It is an interesting topic, and the paper presents a general theory with minimal assumptions, enabling the explanation of abrupt shifts during the sampling phase across different modeling paradigms and data modalities. The writing is clear, and the definition of critical windows based on sub-mixtures, along with the discussion on hierarchical sampling, is engaging.

**Claims And Evidence:**

1. If the reverse process is deterministic, such as an ODE, or includes additional conditions, such as text-to-image, does this theoretical framework still apply?

**Essential References Not Discussed:**

N/A

**Experimental Designs Or Analyses:**

2. Section 4 presents some case studies. Could you provide further experimental results to verify the accuracy of the computed critical windows from the theoretical analysis?

**Methods And Evaluation Criteria:**

N/A

**Other Comments Or Suggestions:**

N/A

**Other Strengths And Weaknesses:**

N/A

**Questions For Authors:**

N/A

**Relation To Broader Scientific Literature:**

This paper proposes a unified and concise theoretical framework that explains the critical windows phenomenon observed in autoregressive and diffusion models in previous studies.

**Theoretical Claims:**

I have checked the proof of Theorem 3.1 and found no additional issues.

---

> ### Author Rebuttal · Authors · 2025-04-01
>
> We would like to thank the reviewer for their time and thoughtful comments. We were glad to hear that you found our writing clear and engaging and our theory interesting.
>
>
> * *“If the reverse process is deterministic, such as an ODE, or includes additional conditions, such as text-to-image, does this theoretical framework still apply?”*
>
>
> If the reverse process is deterministic, then there is no notion of a critical window under our framework. The initial position at the start of sampling completely characterizes the final image. For example, given a fixed piece of text and language model, truncating it anywhere in the model’s response and resampling at temperature $0$ would yield the same final text. This means that the probability it would yield the same answer as the original generation would be 0. We view this as a fruitful future direction of work to extend our framework to deterministic samplers.
>
>
> * *“Section 4 presents some case studies. Could you provide further experimental results to verify the accuracy of the computed critical windows from the theoretical analysis?”*
>
>
> We would like to highlight that **many of the case studies in Section 4** are accompanied by  experiments either in the appendix or in the existing literature:
> * Li and Chen 2024 confirmed that critical windows for mixtures of Gaussians matched with experiments.
>
> * A critical window for the all-or-nothing phenomenon in sparse linear regression can be seen in Figure 2 of Reeves et al. 2019
>
> * The jailbreak critical windows are demonstrated in previous literature, e.g. Haize Labs 2024, and are reproduced in our Appendix F.1.
>
> * In the final revision, we will mention these experiments along with these examples or present some as well as a diagram for a discrete diffusion model that is a mixture of delta measures.
>
>
> Thank you again for your time in reviewing the paper and providing much helpful feedback. If we have addressed your concerns about the paper, we hope you consider raising our score.
>
> Haize Labs. (2024). *A trivial jailbreak against LLaMA 3.* https://github.com/haizelabs/llama3-jailbreak
>
> Li, M., & Chen, S. (2024). Critical windows: Non-asymptotic theory for feature emergence in diffusion models. arXiv preprint arXiv:2403.01633.
>
> Reeves, G., Xu, J. & Zadik, I. (2019). The All-or-Nothing Phenomenon in Sparse Linear Regression. *Proceedings of the Thirty-Second Conference on Learning Theory in Proceedings of Machine Learning Research* 99:2652-2663.

---

### Official Review · Reviewer_TMA6 · 2025-03-10

**Overall Recommendation:** 5

**Summary:**

The authors present a paper that explores critical windows in generative models.  Their paper is heavily theoretical and they propose an understanding that can be applied to a wide range of models.

**Claims And Evidence:**

yes

**Essential References Not Discussed:**

no

**Experimental Designs Or Analyses:**

NA

**Methods And Evaluation Criteria:**

yes

**Other Comments Or Suggestions:**

undefined terms in the abstract (jailbreak), though this is defined quite early in the introduction

**Other Strengths And Weaknesses:**

First, I will admit to being somewhat biased in favour of strong theoretical contributions, but this paper stands out an exceptionally well written, instructional and informative example.  While the underlying theme of the paper is theoretical, the authors bring their contribution to a real issue in diffusion and LLMs.  I also appreciate the included clear examples adding context to the theoretical formulation.

**Questions For Authors:**

NA

**Relation To Broader Scientific Literature:**

I believe this paper will have broad appeal to the machine learning community

**Theoretical Claims:**

I did not check the accuracy of the proofs

---

> ### Author Rebuttal · Authors · 2025-04-01
>
> We thank the reviewer for their kind comments and strong recommendation. We will modify the abstract to say ''hacks’’ instead of ''jailbreaks.’’

---

> > ### Comment · Reviewer_TMA6 · 2025-04-07
> >
> > The system seems to require a rebuttal comment.  Nothing new added here

---

### Official Review · Reviewer_HCM5 · 2025-03-14

**Overall Recommendation:** 4

**Summary:**

The paper theoretically explains sudden behavioral shifts in generative models through critical windows, employing a forward-reverse experiment to study this phenomenon. It introduces Theorem 3.1, which bounds total variation distance to demonstrate that these windows signify transitions between sub-mixtures. The findings are substantiated with examples from diffusion and autoregressive processes.

## update after rebuttal
The authors' response has addressed my concern, so I have raised the score from 3 to 4.

**Claims And Evidence:**

The central claim is supported by Theorem 3.1. Experimental results further confirm the presence of critical windows in generations.

**Essential References Not Discussed:**

I did not notice any missing key references.

**Experimental Designs Or Analyses:**

The results on LLMs clearly demonstrate abrupt changes in output probabilities.

However, several concerns remain:
- Since LLM performance is sensitive to evaluation metrics, a deeper discussion on the robustness of critical windows across different metrics is needed.
- The experimental setup is not rigorously defined or directly validated against the theory, limiting its connection to Theorem 3.1. While TV distance may not be feasible for real distributions, simulations could provide a more direct validation of the theoretical results.

**Methods And Evaluation Criteria:**

The theoretical method uses stochastic localization samplers and mixture models and arrives at a conclusion in TV distance bounds. This is a sensible approach for studying feature localization in generative models.

**Other Comments Or Suggestions:**

See *Experimental Designs Or Analyses*.

**Other Strengths And Weaknesses:**

See *Experimental Designs Or Analyses*.

**Questions For Authors:**

See *Experimental Designs Or Analyses*.

**Relation To Broader Scientific Literature:**

The paper builds on prior work on critical windows in diffusion models (e.g., Sclocchi et al., 2024; Li & Chen, 2024).

**Theoretical Claims:**

I did not check the proofs in the Appendix.

---

> ### Author Rebuttal · Authors · 2025-04-01
>
> We would like to thank the reviewer for their comments. We were glad to hear that you found that our experimental results for LLMs are convincing.
>
> * *“Since LLM performance is sensitive to evaluation metrics, a deeper discussion on the robustness of critical windows across different metrics is needed.”*
>
> In appendix H.1, we include a discussion about the evaluation metrics we used to test our model, and in appendix H.4, we explore the effect of different temperatures. Note that we use standard methods like direct text comparison for multiple choice questions (Lanham et al. 2023) and existing math graders from the literature (Lightman et al. 2023). Given the primary focus on this paper is theoretical, and our experiments are commensurate with these well-cited manuscripts on LLM evaluation and performance, our evaluation metrics and discussion cover a broad range of datasets, models, and other empirical settings that demonstrate the robustness of the critical windows phenomenon.
>
> * *“The experimental setup is not rigorously defined or directly validated against the theory, limiting its connection to Theorem 3.1. While TV distance may not be feasible for real distributions, simulations could provide a more direct validation of the theoretical results.”*
>
> Appendix G describes a **direct validation of our theoretical results for LLMs**, where we actually compute the TV distance to verify our bounds for a real-world model.
>
>
> Thank you again for your comments. If we have addressed your concerns about the paper, we hope you consider raising our score.
>
>
> Lanham et al. (2023). Measuring Faithfulness in Chain-of-Thought Reasoning. *arXiv preprint arXiv:2307.13702*. Retrieved from https://arxiv.org/abs/2307.13702
>
> Lightman et al. (2023). Let's Verify Step by Step. *arXiv preprint arXiv:2305.20050*. Retrieved from https://arxiv.org/abs/2305.20050

---

> > ### Comment · Reviewer_HCM5 · 2025-04-02
> >
> > Thank the authors for the clear response. I will raise the score to 4.

---

> > > ### Author Response · Authors · 2025-04-02
> > >
> > > Thank you so much!

---

### Official Review · Reviewer_n2Rr · 2025-03-24

**Overall Recommendation:** 4

**Summary:**

The paper presents a theory of “critical windows” – intervals in the generation process in which specific features of the generated data emerge – in both diffusion and autoregressive systems. Leveraging the framework of stochastic localization, the authors rigorously characterize when such windows appear. The theory is applied to several scenarios, including diffusion of Gaussian Mixture Models, jailbreaks in LLMs, a minimal model of problem-solving, and in-context learning. It also considers hierarchical distributions, where a hierarchy of critical windows separate different subpopulations. Finally, experiments demonstrate critical windows in LLM solving reasoning tasks.

## Update after rebuttal

I maintain my positive assessment of this work.

**Claims And Evidence:**

The paper is primarily theoretical and rigorously supports its claims.

**Essential References Not Discussed:**

None that I identified.

**Experimental Designs Or Analyses:**

Experiments are well-executed.

**Methods And Evaluation Criteria:**

N/A

**Other Comments Or Suggestions:**

- The plots in Figure 1, taken from related work, are not explained and are hard to read/understand, especially at that point of the introduction. Personally, I don’t see the necessity of including such a figure. I’d encourage the authors either to remove it or – in case they wish to keep it – to make it larger (use vectorized graphics) and explain the content.
- Is the formulation of autoregressive systems as a stochastic localization sampler a novel contribution of the work? If so, I would suggest highlighting it more. Otherwise, the paper should cite previous work showing it.
- L132 (right column): “The can be understood”?
- L144 (right column): That’s true only in the case of Gaussian diffusion.
- The running title is still the ICML template default and should be updated.

**Other Strengths And Weaknesses:**

**Other strengths**
- The paper provides an interesting and rigorous unifying theoretical framework for critical windows in both diffusion and autoregressive models.

**Other weaknesses**
- Section 1.1. “Our contributions” is quite unclear. In particular, I think it lacks a clear and comprehensive list of the contributions of the paper, especially the theoretical ones. It briefly mentions “bounds” (On what? Obtained how? In which framework?). Moreover, is “Generality” really a contribution of the paper? It also mixes theoretical and empirical contributions. Can the authors give a more standard list of contributions, briefly explaining the setting, the obtained insights, and only then the experimental results? The paper is rather dense in content, so I think it would greatly benefit from a clearer outline of contributions. On a side note, also the abstract is not fully informative of the paper's content.

**Questions For Authors:**

- Can you please elaborate more on the complexity of hierarchies learned by diffusion vs autoregressive models, as speculated at the end of Sec. 5? Don’t the two results refer to different data models/distributions?

**Relation To Broader Scientific Literature:**

The paper follows a recent rich literature on critical windows and phase transition in generative diffusion models. In particular, it extends the rigorous results of Li & Chen (2024) to general stochastic localization samplers, relaxing several technical assumptions and including autoregressive systems, and tightening the bounds in the case of Gaussian diffusion.

**Theoretical Claims:**

I checked the validity of the main result (Theorem 3.1).

---

> ### Author Rebuttal · Authors · 2025-04-01
>
> We would like to thank the reviewer for their time and thoughtful comments, especially with respect to the contribution section and the exposition of our theory. We were glad to hear that you found that our rigorous unifying framework was interesting and that our experiments were well-executed.
>
> # Contributions
>
> * *“Can the authors give a more standard list of contributions, briefly explaining the setting, the obtained insights, and only then the experimental results?"*
>
> We will revise the contributions in the final version, separating into clear “theoretical” and “empirical” sections for clarity. On the theoretical side, we will specify that, unlike existing frameworks, our theory **applies across all generative models and data distributions** represented by the stochastic localization framework, as accomplished by Theorem 3.1 and Definition 3.3. Moreover, the theoretical “bounds”, which we clarify as predictions for the location of critical windows, are more precise than that in Li and Chen 2024.
>
> We also highlight applications of our general theory into specific but important contexts as contributions: for example, we can now **compute critical windows in many different contexts** (discrete diffusion, in-context learning, statistical inference), **unlike existing work (Section 4)**.  We will also specify a novel result for hierarchically structured data, where, if the learned sampler and true model are based on the same localization sampler and the learned sampler is good, then they have the same hierarchical structure (Corollary 5.3).
>
> We will add a section that better explains our framework intuitively as well.
>
> * *“It briefly mentions “bounds””*
>
> By this we mean a comparison between the computations of the locations for critical windows for Gaussian diffusion in Li and Chen 2024 versus this paper (Theorem 3.1). They were only able to control the total variation by epsilon times a factor polynomial with the dimension $d$. We were able to upper bound the total variation by epsilon times a constant, and our theorem **improves on their results by a factor that grows polynomially with $d$**. We will clarify this in our contributions.
>
> * *“is “Generality” really a contribution of the paper?”*
>
> By generality, we mean that our framework applies to all stochastic localization samplers and models of data, not just the Gaussian diffusions and the toy models of data considered before in the literature. We view this ability of our **unifying framework** to explain critical windows across so many different contexts as a major contribution of our work.
>
> * *“On a side note, also the abstract is not fully informative of the paper's content.”*
>
> We will synthesize the background in the abstract and better describe our contributions.
>
> # Other Comments or Suggestions
>
> * *“The plots in Figure 1 …  are hard to read”*
>
> In the final revision, Figure 1 will only include three examples that will be explained: the Georgiev et al. 2023 critical window, the prefill attack from Haize Labs 2024, and Phi-4 critical tokens in Abdin et al. 2024.
>
> * *“That’s true only in the case of Gaussian diffusion.”*
>
> While the initial applications of stochastic localization (Eldan 2013; 2020) were Gaussian diffusion, extensions of stochastic localization by Montanari 2023 apply to a broader family of generative models, including discrete diffusion models (Example B.2 and Section 4.3. of Montanari 2023).
>
> * *“Is the formulation of autoregressive systems as a stochastic localization sampler a novel contribution of the work?"*
>
> This was first presented in Montanari 2023. In Section 2.1 and 2.2, we explicitly cite this work. and we will modify the text to cite it in Appendix B when we instantiate language models within this framework.
>
> * *“Can you please elaborate more on the complexity of hierarchies learned by diffusion vs autoregressive models, as speculated at the end of Sec. 5?”*
>
> We view the dimension of a diffusion model as the dimension of the underlying space, $d$ in $\mathbb{R}^d$, and the dimension of an autoregressive model as the length of its context, $T$ in $\mathcal{A}^T$. We simply pointed out that the hierarchy depth was $O(\ln d)$ for a continuous diffusion example and $\Omega(T)$ for an autoregressive example. While they refer to different modalities, we wanted to highlight this vast difference between how hierarchy depth can vary with the dimension. In the final version, we will explain this more clearly.
>
>
> Thank you again for your time in reviewing the paper and providing much helpful feedback. If we have addressed your concerns about the paper, we hope you consider raising our score.
>
> Eldan, R. Thin shell implies spectral gap up to polylog via a stochastic localization scheme. *Geometric and Functional Analysis*, 23(2):532–569, 2013.
>
> Eldan, R. Taming correlations through entropy-efficient measure decompositions with applications to mean-field approximation. *Probability Theory and Related Fields*, 176(3-4):737–755, 2020.

---

> > ### Comment · Reviewer_n2Rr · 2025-04-08
> >
> > I thank the authors for their answers. I maintain my positive assessment of this work.

---

### Official Review · Reviewer_D7NV · 2025-03-26

**Overall Recommendation:** 4

**Summary:**

The paper investigates "critical windows" in generative models—brief intervals during the generation process in which features of the final output are determined. The authors introduce a general theoretical framework based on stochastic localization samplers, a class that includes both diffusion models and autoregressive models as special cases. Their data model assumes samples are drawn from a mixture distribution, with sub-mixtures corresponding to specific features. The core theoretical contribution involves analyzing forward-reverse experiments to identify critical windows: time intervals during which the inversion of a noised observation yields a distribution localized on a sub-mixture, corresponding to the emergence of a feature.

The authors instantiate their theory with examples such as Gaussian mixture models under diffusion and stylized settings modeling jailbreaks, math reasoning, and in-context learning in large language models. They then extend the theory to handle hierarchical mixture models, where modes are recursively nested. Finally, they perform experiments with large language models showing the presence of critical windows during their generation and that these windows are more likely to occur when the model outputs incorrect answers.

## Update after rebuttal
The paper is technically sound, offers some unifying perspectives, and presents interesting experiments on LLMs.
While I still find the predictive power of the framework in empirical settings somewhat limited, the authors have addressed most of my concerns. I have therefore raised my score to recommend acceptance.

**Claims And Evidence:**

The authors claim that their theory applies to a wide class of generative models (both diffusion and autoregressive), that it improves upon prior theoretical results, and that it avoids strong distributional assumptions.
The first part of their claims is well supported. However, I think that the claim that their theory requires "no distributional assumptions" is overstated. In fact, it relies on having data from a mixture model, and it is unclear how to apply it to more complex data structures. Moreover, as the authors acknowledge in Remark 3.2, there are mixture distributions and samplers for which their bounds may be vacuous. Therefore, the applicability of the theory crucially depends on the considered data structure.

On the empirical side, the reported experiments with LLMs provide evidence that critical windows can be identified in their generative process.

**Essential References Not Discussed:**

The essential scientific literature is cited.

**Experimental Designs Or Analyses:**

The experimental designs are sound for the considered tasks. The observation of the existence of critical windows in many LLMs tasks is interesting on its own. However, I am not sure about the connection between the experiments and the proposed theory: is there some qualitative phenomenon we can predict from the theory (e.g., existence or not of critical windows, their width, etc.) that can be then verified in the experimental data?

**Methods And Evaluation Criteria:**

Yes, the methods and evaluation criteria make sense.

**Other Comments Or Suggestions:**

- Example 4.2: $S_{after}$ should be $\{\mu_i\}$

**Other Strengths And Weaknesses:**

Strengths
- The theoretical framework is general and draws connections between stochastic localization and different generative models, such as diffusion and autoregressive models.
- The identification of critical windows in LLM tasks and how they correlate with accuracy is interesting.

Weaknesses
- The theory relies on a specific structure of the data distribution and its features, and the limit of validity of this modeling assumption should be better clarified.
- The connection between experiments and theory is not very compelling.

**Questions For Authors:**

Can the authors clarify how general the data distribution assumption and the theoretical results are?
It seems that the width of the critical windows varies significantly according to the considered model. Is there an intuition about when to expect sharp critical windows?
In the experiments, the presence or absence of critical windows seems to depend strongly on the starting data. Can you elaborate more on that?

**Relation To Broader Scientific Literature:**

The paper improves previous theoretical results on critical windows for diffusion of mixtures of log-concave distributions [Li&Chen 2024].
The studied phenomena are connected to similar studies of diffusion models from a statistical physics perspective, which focus more on specific data models [Raya&Ambrogioni2023, Sclocchi et al. 2024, 2025; Biroli et al. 2024].
The paper connects these ideas of critical windows in diffusion models with recent observations in Jailbreaks and chain-of-thought in LLMs.

**Theoretical Claims:**

I checked the correctness of the proof of the main theorem, which is sound and logically well-organized.

---

> ### Author Rebuttal · Authors · 2025-04-01
>
> We would like to thank the reviewer for their time and thoughtful comments, especially with respect to our theory’s assumptions and the relationship between our experiments and theory. We are glad that you found that the generality of our theory and LLM reasoning experiments interesting.
>
> **Modeling assumptions**
> * *“The claim that their theory requires "no distributional assumptions" is overstated.. it relies on having data from a mixture model… Can the authors clarify how general the data distribution assumption and the theoretical results are?”*
>
> In the final revision, we will rephrase “no distributional assumptions” to “very few distributional assumptions.” The mixture model assumption is **extremely mild**; any partition of the outputs from a generative model defines a mixture model, where the classes are given by the different partitions. For example, for a list of outputs of a language model, we can split them into labeled groups such as {correct answer, incorrect answer}, {safe answer, unsafe answer}, etc. The ability to attach these labels and partition into subpopulations is broadly applicable to the datasets, which we also make use of in our experiments.
>
> See Remark 2.4 as well for a re-emphasis of the generality of the mixture model assumption.
>
> * *“It seems that the width of the critical windows varies significantly according to the considered model... the applicability of the theory crucially depends on the considered data structure... limit of validity of this modeling assumption should be better clarified.”*
>
> We view one of our main contributions as offering a **unifying framework** to distill the phenomenon of critical windows to very general facts about the data distribution, so that their **presence/absence or width** depends only on computations with the data model and forward process. This is a major strength compared to extant literature, which only discusses critical windows for **particular data distributions**. In the final revision of this paper, we will clarify that our bounds and the narrowness of our critical windows are affected by the specifics of the data distribution.
>
> * *“there are mixture distributions and samplers for which their bounds may be vacuous.”*
>
> In Sec. 4 and 5, we verify that our bounds are non-vacuous in many contexts.
>
> * *"Is there an intuition about when to expect sharp critical windows?"*
>
> In Ex. 4.3, we mention an example providing general intuition when critical windows can be sharp, i.e. when a few tokens are very unlikely under one mode compared to the other. In general, we expect sharp critical windows when it only takes a few steps from the forward process to erase the differences between $S_{\textrm{before}}$ and $S_{\textrm{after}}$. This could happen if the data has a multi-scale hierarchical structure (Definition 5.1), where a feature is decided in a narrow intermediate band of the tree.
>
> **Experiments**
> * *“Is there some qualitative phenomenon we can predict from the theory (e.g., existence or not of critical windows, their width, etc.) that can be then verified in the experimental data?”*
>
> We highlight **several predictions of our theory** verified by experiments:
> * We provide structured output experiments where our predictions for the location of critical windows match experiments (Fig. 5, App. G).
>
> * We predict that prefill jailbreaks yield narrow critical windows, because the probability that a model agrees to a harmful request in the first few tokens but refuses in the end is low (Ex. 4.3). This is consistent with (Haize Labs 2024b) and Fig. 4a.
>
> For LLM reasoning experiments, our only claim is that critical windows coincide with reasoning mistakes (Table 1). In the final revision, we will make which aspects of our theory that the experiments verify more clear.
>
> We note other works verifying theory with experiments: Li and Chen 2024 showed that positions of theoretical computations of critical windows matched up with experiments for Gaussian mixtures, and Biroli et al. 2024 showed a measure of the separation (size of principal component) between classes predicts real-life critical windows for diffusion.
>
> * *“In the experiments, the presence or absence of critical windows seems to depend strongly on the starting data. Can you elaborate more on that?”*
>
> We agree that the specifics of the starting point could affect the location or presence of the critical window (Fig. 7). One explanation is that sometimes certain parts of the solution are more important to the final answer. For example, the bolded critical window in Figure 3 occurs at the point where the model finds the correct formula is crucial to solving the problem. In other instances, no particular part of the text could be crucial to the answer. We will clarify this in the final revision.
>
> Thank you again for your time in reviewing the paper and providing much helpful feedback. If we have addressed your concerns about the paper, we hope you consider raising our score.

---

### Decision · Program_Chairs · 2025-05-01

**Decision:**

Accept (oral)

**Comment:**

The paper studies the emergence of “critical windows” in generative models, namely locations in a generative process where some underlying features are determined. While this has been studied previously in diffusion models under strong distributional assumptions, the theory presented here refines existing bounds and applies more broadly to different classes of generative models and with very mild assumptions on the data distribution. The reviewers agree that this is a solid contribution and are all in favor of acceptance. Please include the proposed clarifications in the final version.